# Messaging Protocols for IoT Systems—A Pragmatic Comparison

**DOI:** 10.3390/s21206904

**Published:** 2021-10-18

**Authors:** Jacek Wytrębowicz, Krzysztof Cabaj, Jerzy Krawiec

**Affiliations:** 1Institute of Computer Science, Warsaw University of Technology, ul. Nowowiejska 15/19, 00-665 Warsaw, Poland; krzysztof.cabaj@pw.edu.pl; 2Institute of Production Systems Organisation, Warsaw University of Technology, ul. Narbutta 85, 02-524 Warsaw, Poland; jerzy.krawiec@pw.edu.pl

**Keywords:** internet of things, networking protocols, messaging protocols, IoT systems

## Abstract

There are a dozen messaging protocols proposed for IoT systems. Choosing one for a new design is complicated, and a non-optimal selection can result in slower development and higher design costs. This paper aims to help select appropriate protocols, considering IoT applications’ specificity and communication requirements. We have identified the protocol features that are significant for the design and operation of IoT systems. This paper gives a substantial comparison of the protocols using the features and is based on a thorough analysis of the protocol specifications. The results contain an assessment of the suitability of the protocols for the defined types of IoT devices and the identified communication purposes. We conclude the comparison with some recommendations of the protocol selection and usage.

## 1. Introduction

IoT systems are designed to support very different applications and to work in various environments. That explains the enormous numbers of available components, software libraries, and tools [1,2] along with communication standards [3]. Moreover, the competition on the market and the observed rapid progress of hardware and software technologies make designers struggle with abundance.

This article analyses messaging protocols available for IoT system designers. The protocols simplify communication between application processes. They separate the processes from transport and network protocols. Some provide specific functionalities like Quality of Services (QoS), information tagging, or middlebox services useful for information caching and routing. Most of the existing messaging protocols are available as open-source programming libraries. Many of them are included in IoT platforms, which are out-of-the-box solutions that encapsulate significant parts of an end-to-end IoT system. The problem is that the protocols are numerous, and choosing the appropriate one for a given solution is not a trivial task. We can mention MQTT, CoAP, STOMP, XMPP, WAMP, AMQP, DDS, OPC UA, LwM2M, Weave, and HomeKit. Moreover, an IoT system designer can select one of the existing communication programming libraries with built-in non-standard messaging protocols, e.g., the YAMI4 (http://www.inspirel.com/yami4/, accessed on 13 October 2021) or ZeroMQ (https://zeromq.org, accessed on 13 October 2021) middleware.

We can design an IoT system without messaging protocols, building application programs on top of a general-purpose transport protocol, e.g., UDP, TCP, or QUIC. Such an approach is good if the system is not very complex and should work inside an enterprise network. If some IoT devices should be accessible from web browsers, they should have HTTP servers on-board, and HTTP could be the only needed application layer protocol. According to the application needs, HTTP should be secured by TLS, and such settlement is called HTTPS. The advantage of using HTTP is that most firewalls do not hinder communication and it is easy to find experienced programmers. The advantage makes that the protocol is widely used instead of the mentioned transport protocols by new network applications. The WebSocket mechanism (RFC 6455), conceived to support better interactivity between web browsers and web servers, is used as a transport protocol by many of today’s applications. One WebSocket server can support many application protocols at the same time. Most of the messaging protocols have defined profiles for running over WebSocket. Both HTTP and WebSocket can and are used directly by IoT applications. However, most IoT applications profit from messaging protocols, which provide additional functionality, non-offered by the transport protocols, HTTP and WebSocket. We are going to compare those additional functionalities.

An IoT system designer can use e-mail protocols (e.g., SMTP, IMAP, POP3) for message exchange; he or she can even use social media as a transport layer platform (e.g., Twitter). We decided to exclude them from the comparison. The principal aim of e-mail is to serve a human at least at the recipient end of the communication. Moreover, today’s e-mail services are overwhelmed by spam. We cannot consider them as a recommended solution for communication between IoT devices and services. A solution based on a social media can be considered a simple transport without specific functionalities expected in IoT communication.

This work aims to provide a comprehensive and pragmatic comparison of messaging protocols considered for new IoT projects to help designers choose. The following aspects are analysed: protocol functionality, the complexity of documentation (which reflects the cost of learning), and the universality of use. We do not analyse security aspects such as authentication, confidentiality, and integrity, as they should be implemented in the lower layers of the protocol stack. However, possible authorisation mechanisms are included in the functional comparison. Even though many articles compare protocols for IoT systems, most of them focus on performance analysis of selected implementations, and the comparisons are often limited to a few selected protocols.

The main contributions of the paper are as follows:description of a rich set of messaging protocols, distinguishing their unique features,thorough and methodical comparison of the protocols,pragmatic evaluations and recommendations for IoT system designers.

The rest of this paper is organised as follows. Section 2 analyses the communication requirements of modern IoT systems. Section 3 defines the comparison methodology used in this work. Section 4 gives succinct descriptions of MQTT, MQTT-SN, CoAP, STOMP, XMPP, WAMP, AMQP, DDS, OPC UA, LwM2M, and proprietary protocols for smart-home devices. Section 5 provides the comparative analysis of the protocols presenting in three subsections: comparison of the protocol functionalities, maturity and complexity, and suitability for selected types of IoT devices with suitability for basic communication purposes. Section 6 presents related works. The paper concludes in Section 7, where some pragmatic recommendations are given.

This paper does not describe nor analyses the Common Industrial Protocol (CIP). Either CIP is considered a solution for the Industrial Internet of Things (IIoT), it is not a universal messaging protocol that can be an alternative for any mentioned above. CIP is a solution for integration and inter-communication with different types of field networks, i.e., Ethernet/IP/IP, ControlNet, DeviceNet, CompoNet, and Modbus.

## 2. Communication Requirements of IoT Systems

An IoT system can contain subnetworks of different kinds of devices. There are constrained devices, i.e., with limited computation and memory resources due to price or energy limitations. The communication protocols installed on such devices should have a small memory footprint. Moreover, the protocols should minimise the communication overhead to save energy.

Some devices can be stationary or moving. Some mobile devices do not transfer data in motion, while others do. The device mobility causes that the IP address can change and cannot serve as the device identifier. A consequence of mobility can be a temporal gap in communication. The mobility can be easily achieved using services of wide-area radio networks, e.g., LTE, Sigfox, LoRa, which are paid per-use. Furthermore, some devices go to sleeping mode due to energy saving or their expected functionalities, e.g., a need for data sensing according to a given schedule. A proxy or broker, which could provide data when the IoT devices are unavailable, is introduced in such a case.

We can compose a device-level IoT platform, i.e., a middleware to be installed on a group of IoT devices that helps rapid development of a new system. A selected messaging protocol or a set of protocols is a part of the device-level IoT platform. Such middleware should be optimised for specific communication needs of a given IoT device type. We can distinguish four types of such devices, i.e., the following:Constrained devices.Unconstrained devices connected to the internet via a pay-per-use plan or a flat-rate plan with significant limits on the volume of transferred data.Unconstrained devices being not always online.Unconstrained, online devices anytime accessible by applications or services.

A cloud or edge server is unconstrained and always online. It can interact with several IoT subnetworks or devices belonging to any of the above types. A communication middleware installed on the server should respect the limitation imposed by the devices. Moreover, the middleware should support the needs of the designed application. The middleware can contain several messaging protocols to enable communication with different IoT subnetworks. It is also possible to install in the network middleboxes that are messages brokers and operate several protocols to communicate with different IoT devices. We will analyse the suitability of different messaging protocols for the above-mentioned types of IoT devices.

An IoT system usually includes many processes and services that run on cloud, fog, or edge servers; we will call them further internet actors. We will consider their communication with processes and services that run on IoT devices; we will call them further IoT actors. The communication needs of the internet actors can vary depending on performed functionalities. We can distinguish a few elementary types of communication purposes:Device discovery and configuration—the IoT actor calls service from an internet actor infrequently.Data acquisition—the IoT actor sends data repeatedly.Data querying—the IoT actor sends data only on request.Notification and alarm analysing—the IoT actor sends priority data infrequently.Command dispatching—the IoT actor receives commands.Process control—messages are sent as responses to sensed data, or some feedback is expected after control commands; the internet actor repeatedly calls service from an IoT actor.Opportunistic peer-to-peer data exchange—the IoT actors directly share some information or services.

There can be other communication purposes, like software updates, system health analysis, security analysis, but their communication schemas belong to the above set. We will analyse the suitability of different messaging protocols for all of the above schemas.

The messaging protocols have diverse built-in mechanisms that simplify the design of different communication patterns (paradigms). Distant processes can play some communication roles depending on performed tasks. A process can act as a data producer, consumer, message broker, RPC (remote procedure call) service, callee, or dealer. Data can be delivered on request (the client-server approach) or just after production (the publish-subscribe approach). If the communication channel is slow, then the latter approach is more efficient. The publish-subscribe approach can apply a queue or a mailbox. The queue serves to process incoming messages in the reception order. The mailbox serves to process incoming messages in any order, e.g., depending on the sender or the message subject (topic). The data producer can be a server that sends data on request or a publisher that sends data on subscription, and the data consumer can be a client or a subscriber.

There are two asymmetric communication patterns, master-slave and client-server. If one node manages the message exchange, then we have master-slave communication. If one node serves requests from many clients, then we have client-server communication. Symmetric (peer-to-peer) communication is possible if the network nodes can play the same and complementary roles—the node is server and client or publisher and subscriber or RPC caller and callee simultaneously. There are two peer-to-peer patterns: 1-to-1 and N-to-N. The second is possible if the communication protocol enables the delivery of a once sent message or an RPC call to many nodes.

If a given protocol uses connection-oriented transport like TCP, the listening party is called a server, and the connection initiation party is called a client. It is due to the TCP convention, regardless of the role the application process performs. As a result, the commonly used terminology of communication paradigms is a bit inconsistent. The term client-server means the pattern with listening and connecting parties, and also can suggest the architecture pattern with many client nodes that request some services from the server node.

More complex communication patterns are applied to large systems with nodes that can dynamically appear, be available or not. We can distinguish four architectures:With message broker—the broker interconnects many publishers and subscribers. It can temporally store messages and control access rights. Implementation of a broker can be centralised or distributed.With RPC dealer—the dealer interconnects many RPC callers and callees. It can control access rights and manage the bounds between end-parties. Implementation of a dealer can be centralised or distributed.With message router—the router selects the destination or finds a path to the destination if several routers belong to the system. The router can process annotations attached to the forwarded message. That paradigm enables the high scalability of the system.With message server—the server can do some message processing, e.g., it can aggregate them. Moreover, the server can act as a broker or a message router.

A messaging protocol can support in a way one of the complex patterns. However, it does not define the internals of the intermediary nodes. We will compare how the protocols fit the communication paradigms. If a protocol does not match the needed pattern, the programmer has some more coding to adapt the protocol to the application needs. For example, a client can push produced data to a server, pull some data for consumption, send an RPC call, or ask about a call to its procedure.

## 3. Comparison Methodology

An attempt to compare protocols that have been in use for years, that evolve, that have been analysed in many ways is challenging. Any performance aspect (e.g., related to energy consumption or time response) depends on more factors, not only the selected messaging protocol. The way of its usage and the choice of lower-level protocols are significant. Moreover, optimisation of its implementation and selected hardware/software environment influence efficiency. Results of performance comparison for one application scenario must not hold in a different scenario. Changing a library that implements a given protocol may result in performance (in terms of latency, memory, and CPU consumption) improvement or deterioration by one order of magnitude; see, for example, the Iglesias-Urkia’s et al. analysis of CoAP implementations [4]. For all these reasons, we give up the performance comparison.

We intend to help engineers select the protocol for new designs. Thus, we provide analysis and comparison of supported communication paradigms, functionalities, maturity, complexity, suitability for the four defined above types of IoT devices, and suitability for the seven defined communication purposes. All these aspects are described in the next section, which introduces each analysed protocol. The comparison chapter presents values of identified below protocol features, giving some tables and figures for better readability. The comparison is based on a thorough study of the available specifications of the protocols and their extensions.

The formation of any protocol has been led by an objective that reflected the intended application class, assuming one or more communication paradigms. The features related to such objectives are the roles defined for communicating nodes, the architecture, a built-in discovery mechanism, real-time support, and additional features. The additional features can characterise programming style, e.g., object-oriented or RESTful.

We split the protocol functionalities into messaging and transport features. The messaging features are constraints for maximum payload size, possible data representations, support for labelling, metadata and transaction processing. The transport features are possible transport protocols, security properties, QoS, prioritisation, message addressing, and filtering.

These supported communication paradigms and functionalities are more important than the novelty of a protocol. The protocol maturity and stability are of value for a large or industrial project. For that reason, we provide information about the first and the recent dates of protocol issues and the publication dates of their newest extensions.

We try to assess the complexity of the protocols. The complexity affects the memory footprint of an implementation; hence, it should be considered for systems with constrained devices. Moreover, the complexity of a protocol impacts learning time, so project cost. We assess the learning difficulty by comparing the volume of protocol specifications counted in page numbers. The programming difficulty can be assessed by comparing the number of PDU (Protocol Data Unit) types and protocol elements. PDU is a message (a data structure) exchanged and processed by distant protocol instances. The protocol elements are different fields in the fixed and variable headers and properties specific for every PDU type. The number of PDU types reflects how many different operations an application can call. The number of elements reflects how many parameters a programmer can set.

## 4. Characteristic of Selected Messaging Protocols

We present in this chapter a brief review of the selected protocols. We aim to show their most important functionalities and features needed for their comparison. In the following sections, we describe MQTT, MQTT-SN, CoAP, STOMP, XMPP, WAMP, AMQP, DDS, OPC UA, LwM2M and proprietary protocols for smart-home devices.

### 4.1. MQTT

MQTT (Message Queuing Telemetry Transport) is an open standard maintained by OASIS [5]. The MQTT paradigm is publish-subscribe with a broker. Publishers and subscribers act as clients, while the broker acts as a server. The broker is an intermediary node that relays messages accordingly to their topics. The topics are organised in a hierarchical structure. The broker can delete a message sent to all subscribers of the given topic; it can also delete a message if there is no subscriber. An IoT device and a network service or process can embed a publisher, a subscriber, or both. The broker can be installed on a cloud or edge server or on an IoT gate that separates a network of devices from the internet. A server supporting the broker can be at the same time a publisher and subscriber to another broker—what enables hierarchical and scalable deployments.

There are three QoS modes of data delivery. QoS0—at most once—it is a best-effort delivery. QoS1—at least once—duplicates can occur. QoS2—exactly once—a reliable delivery. The higher the QoS type, the more PDUs are exchanged. A subscription can be durable or non-durable. The durable subscription instructs the broker to store messages for a given client when disconnected, e.g., in a sleeping mode. Moreover, a publisher may instruct the broker to retain messages even if there are no registered subscribers for the topic. In such a case, a subsequent message will overwrite the previous one, and an appearing subscriber will get the more recent value of the topic.

The recent MQTT version 5.0 was published in 2019 (137 pages). MQTT has evolved since 1999. Moreover, since 2017, some work has been led to define the TLS profile for MQTT [6] (33 pages). The profile enables authorising a connection between a client and the broker and protecting resources identified by topic names. An authorisation server provides tokens to the clients. The token proves access right to the broker for publishing or subscribing on a given topic.

Some solutions to secure MQTT were proposed in the past. For example, in 2014, Neisse et al. [7] integrated their Model-based Security Toolkit with MQTT to support security and privacy requirements. In 2015, Singh et al. [8] defined a security extension for MQTT and MQTT-SN called Secure MQTT (SMQTT). The extension is based on the lightweight Elliptic Curve Cryptography and allows broadcasting encrypted messages to multiple nodes. However, the mentioned solutions were not included in the latest MQTT version from 2019, which introduces, among others, an enhanced authentication method. The method is commonly used to carry the SASL mechanism, but it is not constrained to SASL, and others like Kerberos can be applied. The MQTT specification strongly recommends using TLS for securing message exchange with the broker.

MQTT works on top of TCP or TLS, or WebSocket. The MQTT standard defines 14 PDU types. The PDU length can range from 2 B to 256 MiB, and its header length can vary from 2 B to 5 B. The content of the published payload is application-specific. The UTF-8 text string codes the topic, and the payload can be a byte or UTF-8 text string. The MQTT standard defines several protocol fields in the fixed and variable headers and many properties specific for every PDU type. The application programmer can set the values of these protocol elements; only some of them are set by the protocol. Their total number is 99, which characterise the complexity of the protocol.

MQTT was designed to minimise the complexity of a client implementation; for example, its C library size is in the range of 30 KiB and Java in the range of 64 KiB. However, the broker implementation should be robust. Redundancy can be needed to eliminate the risk of a single point of failure. When high scalability is needed, a distributed broker implementation should be done in a way. Though, the redundancy and scalability issues are out of the MQTT standard scope. MQTT is supported by many programming libraries, frameworks, and tools associated with learning materials; see https://mqtt.org (accessed on 13 October 2021).

Any IoT platform for unconstrained devices can use MQTT. The upper layer takes responsibility for proper dealing with network availability and the possible cost of data transfer. Some constrained devices can support TCP and non-compressed messages due to the infrequent transfer of short data. However, in general, the MQTT standard is not a good choice for constrained devices.

An IoT device that is only a publisher can be used for data acquisition and notification/alarm analysing. An IoT device that is only a subscriber can be used for the execution of dispatched commands. An IoT device that acts as both publisher and subscriber can be used in device discovery and configuration processes, data querying, and remote controlling processes. However, MQTT is not suitable for the case that serves peer-to-peer communication between neighbouring devices.

MQTT is used in a large variety of IoT applications. There are known industrial MQTT deployments covering hundreds or thousands of topics/subtopics and similar numbers of active terminal nodes. A particular example of MQTT usage is Facebook Messenger. A valuable review of MQTT applications and comparison of various MQTT programming libraries can be found in a recent work by Mishra and Kertesz [9].

### 4.2. MQTT-SN

MQTT-SN (MQTT for Sensor Networks) [10] is a variation of MQTT developed by IBM and accepted by OASIS. It is aimed at constrained devices. It works efficiently over wireless radio links, where low speed and high failure transmission rate is expected, and messages should be short.

MQTT-SN provides the same functionalities standard MQTT does. The main difference is that it can work in non-TCP/IP networks. It was initially developed for running on top of ZigBee, but Bluetooth and UDP are also used. Other differences between MQTT and MQTT-SN can be perceived as extensions. There are new architecture elements, i.e., transparent and aggregating gateways, which facilitate the planning of the range of radio links; moreover, they enable load-sharing. There is defined a discovery procedure of an active server/gateway. Next, a keep-alive procedure that supports sleeping clients—a server/gateway buffers then messages. Additionally, QoS mode (called -1 level) is supported. This mode is intended for very simple terminals that publish only messages on one predefined topic. Such a terminal neither opens nor closes the connection with the broker nor does it receive message acknowledgement. An MQTT-SN broker can serve data exchange only between MQTT-SN clients. However, with the help of the MQTT-SN gateway, any existing MQTT broker can be used, and then communication with a client of any type can occur. Moreover, there are not many MQTT-SN brokers available currently.

The recent MQTT-SN version 1.2 was published in 2013 (28 pages). However, its specification should be considered as an extension of the MQTT standard (137 pages). MQTT evolves since 2007.

MQTT-SN can work on top of any unreliable packet transport protocol. The MQTT-SN standard defines 28 PDU types. The PDU length can vary from 2 B to 64 KiB; however, in some networks, the length can be limited; e.g., in ZigBee, the maximum length is restricted to 60 octets. The PDU header length can vary from 2 B to 4 B. Instead of passing long topic names in messages, their short numbers or names are carried over (1 or 2 B). The published payload is application-specific and carried as a string of bytes. The variable header is simplified. The total number of different fields defined by the MQTT-SN standard is 18.

MQTT-SN extends the use of standard MQTT by effectively supporting communication with constrained devices. Together, the two protocols can back up a device-level IoT platform of any type. However, MQTT-SN does not extend the list of suitable communication schemas. MQTT-SN is not suitable neither for the case that serves peer-to-peer communication between neighbouring devices.

### 4.3. CoAP

CoAP (Constrained Application Protocol) is an IETF standard. All IETF standards and drafts can be accessed on the webpage https://datatracker.ietf.org (accessed on 13 October 2021). RFC 7252 defines the core CoAP specification, and several other RFCs define CoAP extensions. Moreover, a dozen further improvement proposals can be found between active IETF drafts. CoAP can be considered as a specific version of the HTTP protocol designed to communicate with resource-constrained devices. It suits the RESTful programming approach and works according to the client-server paradigm. The server provides resources identified by the URIs. The client communicates with a server with a known, predefined address, although it is possible to profit from the mechanism of dynamic server discovery in local networks. A given device can act as both a server and a client. An extension, defined by RFC 7641, enables the client to subscribe to server resource updates over a period of time, providing the best-effort publish-subscribe communication pattern. CoAP supports a few communication architectures, i.e., peer-to-peer, client-server, master-slave. It can be applied in more complex architectures, too.

CoAP implements two QoS modes, delivery with and without acknowledgement. An acknowledgement can be delayed in order to minimise the number of sent messages. An application designer should be aware that CoAP offers two retransmission parameters to be tuned, i.e., ACK_TIMEOUT and MAX_RETRANSMIT. If the retransmission counter expires, then the delivery is not guaranteed. It is possible to configure CoAP to forward messages to a group of recipients—extensions for group communication are defined by RFC 7390. In turn, RFC 7959 defines a way to pass large blocks of information using short CoAP messages—this feature can be used to update the software or configure the IoT device. Dedicated proxies can process CoAP messages with various functions, such as logging data to the cache memory or representing devices using other protocols. It is possible to deploy a proxy that translates CoAP messages from/to HTTP messages.

The core CoAP specification was published in 2014 (112 pages). There exist its errata from 2017. The first CoAP draft was published in 2009, and we can find open-source libraries written in 2010, 2011, and 2013 according to the draft specifications from those years. Twelve RFCs define extensions and recommendations for CoAP (395 pages in total). The newest was published in 2021, and we can be sure to have more CoAP-related RFCs, seeing the number of active drafts.

UDP and DTLS are the primary protocols for carrying CoAP messages. Moreover, these messages can be transported over other protocols such as TCP, SCTP, WebSocket, or even SMS (Short Message Service from mobile telephony) transport was proposed. CoAP offers 7 PDU types, i.e., 4 request methods (GET, POST, PUT, DELETE) and the response, acknowledgement, reset messages. However, RFC 8132 defines 3 more PDUs for CoAP, i.e., FETCH, PATCH, and iPATCH. The message header is only 4 B long, and the main protocol fields (token, options, payload) are of variable length. The number of all defined fields that can appear in the message frame is 24.

The maximum message length is limited—as a single IP packet should carry the message without fragmentation. Most networks have a limit of 1024 B, but some networks may restrict the length even to 40 B. The payload of a request or response typically carries a representation of a resource. Its format can be any internet media types, e.g., text/plain, application/senml + json, application/cbor.

Any IoT platform for unconstrained or constrained devices can use CoAP. However, in the case of constrained devices, special attention should be directed to how the payload is coded to minimise its volume.

An IoT device being only a client can be used for data acquisition, notification/alarm analysing, and device discovery and configuration. An IoT device being only a server can be used for: the execution of dispatched commands, data querying and remote controlling processes. An IoT device acting both as a client and as a server can be used in any communication schema, including peer-to-peer communication between neighbouring devices.

CoAP can be used in any Web-of-Things application. It is as flexible as HTTP but more suited for device-to-device communication. It is efficient for communication with constrained devices under the condition of thoughtful design and programming.

### 4.4. STOMP

STOMP (Simple/Streaming Text Oriented Messaging Protocol) specification is developed and maintained by the community of programmers (http://stomp.github.io, accessed on 13 October 2021). The STOMP follows the paradigm publish-subscribe with a broker (STOMP server). A client can work as a publisher, a subscriber, or both. Thanks to the server, STOMP can provide one-to-many communication. The server is an intermediary node that relays messages accordingly to the destination value. The STOMP destination plays the same role as the MQTT topic. Several servers can be deployed in a chain that links two clients. The server can modify and add headers to transmitted messages. The protocol supports the client-message_server-client communication pattern. The message server can do some message processing, which is more than just a broker functionality.

STOMP messages have an HTTP-like text syntax. The message consists of the name of the command or response, a set of optional headers composed of the pairs of <attribute, value>, and the body of the message. STOMP enables the transfer of binary content. The content-type header can indicate any media type, and the implicit encoding of the message is UTF-8.

The recent STOMP version 1.2 was published in 2012 (18 pages). The first mention of the protocol on the internet (http://kasparov.skife.org/blog/src/stomp/ttmp-is-named-stomp.html, accessed on 13 October 2021) was in 2005.

STOMP messages are carried over TCP. It is also possible to use WebSocket to avoid problems with firewalls that are blocking most TCP ports. There are 10 PDUs (commands) in STOMP. Commands can be followed by headers (key-value pairs). Only three of them (SEND, MESSAGE, and ERROR) may carry a body. The commands and headers are encoded in UTF-8, which is also recommended for the body. However, any other internet media type can be used and pointed in the header. The maximum message length is implementation dependent; the server may define limits for the body length and the number of headers. The STOMP specification defines 19 headers, so we have 20 protocol elements together with the commands. Any user-defined, application-specific header is allowed.

STOMP is a simple and flexible protocol. It is even possible to execute a communication session with the server using the telnet tool for testing purposes. The destination semantics and reliability of message exchange are left to the server implementation, providing many other functionalities using the STOMP header mechanism. Several known STOMP server and client implementations are listed on the project webpage (https://stomp.github.io/implementations.html, accessed on 13 October 2021).

Any IoT platform for unconstrained devices can use STOMP. Even though some constrained devices can support TCP and non-compressed messages due to the infrequent transfer of short data, STOMP is not a good choice for constrained devices.

Depending on the application’s needs, the functionality of a STOMP server can be reduced or rich; thus, its software can be pretty simple or very complex. Regardless of how simple the software is, it is difficult to find an application where an IoT device should work as a message broker. IoT device being a STOMP client, can be used in any communication schema, excluding peer-to-peer communication between neighbouring devices. However, the STOMP server can be a part of an application server that controls a set of IoT devices—the STOMP clients.

### 4.5. XMPP

XMPP (Extensible Messaging and Presence Protocol) is standardised by IETF. Its basic specification is composed of the following: RFC 6120 defining the core of the protocol, RFC 6121 defining the construction of instant messaging and presence functionality, RFC 7622 defining the address format. A dozen RFCs define various aspects of the protocol operations.

XMPP follows the client-server paradigm. Several XMPP servers can be interconnected. The client connects to the selected server, thus opening itself to other clients connected to this or other servers. XMPP creates a one-way stream to the target. Inside the stream, it sends any number of individual XML phrases (aka stanzas).

A significant feature of XMPP is the ability to transfer the presence status between users. Moreover, XMPP allows persistent streams over constantly open TCP connections, ensuring minimal latency. Although XMPP messages are exchanged according to the client-server paradigm, the communicating users form a peer-to-peer network (with equal rights) from the application point of view. Thanks to XMPP extensions, it is easy to organise communication between users according to any paradigm, i.e., peer-to-peer, client-server, publish-subscribe, message broker. However, the primary communication pattern of the protocol is client-message_router-client. The message router finds a path in the network of XMPP routers, which is more than just a broker functionality.

Each user, human, or process, is assigned an XMPP address (UTF-8 coding), similar to email addresses and consists of three parts: local-part@domain-part/resource-part, e.g., anything@example.com, thisthing@example.com/chamber1. For historical reasons, the address is called Jabber Id. The user can connect to a given server only if he or she has an account registered on that server. The XMPP address format and the principles of server operations simplify a gateway deployment between a given XMPP system and email servers.

The XMPP specification defines a mechanism for establishing connections between the client and server and between servers. Besides, it specifies how to use the StartTLS mechanism to start an encrypted connection on the same port as an unencrypted connection and use SASL (RFC 4422) for user authentication. The main task of the XMPP server is to authenticate clients, maintain their streams, and pass stanzas. Also, the server stores selected data, e.g., a user’s contact list and their status (aka roster). Moreover, it can implement additional communication services, such as multicast and message broker.

The XMPP Standards Foundation (https://xmpp.org, accessed on 13 October 2021) promotes the protocol, develops extension proposals, and standardises them. The foundation carried out some work on XMPP extensions addressed to IoT systems. Several documents have been developed on message compression, sensor networks, service delivery, device discovery, and more. The current status of these documents is withdrawn or deferred. Anyhow, the foundation still works on new extensions to XMPP (https://xmpp.org/extensions, accessed on 13 October 2021).

The three RFCs specifying the XMPP core were published in 2011 and 2015 (352 pages). There exist errata from 2017. The first RFC defining XMPP was published in 2004. The protocol evolved from the Jabber text messaging technology, which appeared in 1999. The first IETF draft specifying XMPP appeared in 2002. Nowadays, eighteen RFCs define extensions and recommendations for XMPP (725 pages in total with the core); we do not count obsoleted RFCs. The newest was published in 2019. There is not any active draft related to XMPP.

XMPP works on top of the TCP protocol. It is also possible to use HTTP or WebSocket to bypass restrictive firewalls. This protocol enables the exchange of XML streams in almost real-time, i.e., with such delays as in a given network occur for transmission within TCP connections. The streaming content is structured in stanzas of 3 types: Message, Presence, and Info/Query. The types can be considered as PDUs of the XMPP protocol. The limit of stanza size is implementation-specific; it should be part of server configuration. The client can send an unbounded number of stanzas over the stream. Any binary data must be base64 encoded before it can be transported. XMPP exchanges XML phrases to manage the stream, the roster, and go through StartTLS or SASL processes. There are 13 such phrases, and they can be considered as PDUs. As a result, we can distinguish 16 PDUs of the XMPP protocol. The number of defined XML elements for which a value can be assigned is 100. Those numbers show the complexity and rich functionality of the protocol.

Any IoT platform for unconstrained devices can use XMPP. Even though some constrained devices can support TCP and non-compressed messages due to the infrequent transfer of short data, XMPP is not a good choice for constrained devices. Setting up a session for each short data exchange makes XMPP inefficient in terms of protocol overhead. Careful application programming is needed in the case of devices connected to the internet via a pay-per-use plan. Keeping long TCP connections could result in unnecessary costs; thus, the XMPP session should be closed immediately after every data transfer or exchange.

Depending on the application’s needs, the functionality of an XMPP server can be reduced or complex. The server is not intended to work on terminal devices. An IoT device being an XMPP client can work in any communication schema, excluding direct peer-to-peer communication between neighbours. However, a peer-to-peer connection between terminal devices via an edge or cloud XMPP server can be achieved.

XMPP is a popular and universal message exchange protocol. It has many extensions, and as a result, it can be found in many very different applications. The main fields of application of this protocol are instant messaging, social networking services, online games, content syndication, and signalling for VoIP/video.

### 4.6. WAMP

WAMP (Web Application Messaging Protocol) is an open protocol [11] maintained by Crossbar.io (https://wamp-proto.org, accessed on 13 October 2021). WAMP is a WebSocket sub-protocol that allows remote procedure calls and publish-subscribe messages to be forwarded. The protocol supports the client-message_router-client and client-RPC_router-client communication patterns.

A client communicating with the WAMP router (server) can register offered procedures to be executed and topics of expected messages. Another client can call registered procedures and send messages on specific topics. The router then acts as a message broker, remote procedure dealer, or both. There is no restriction for an application to play several roles from the following set: callee, caller, publisher, subscriber, dealer, and broker. The WAMP protocol defines communication between client and router. It is recommended to use TLS to ensure communication security. Besides, routers can have realms defined as administrative domains that separate the namespaces of message topics and offered procedures. Access to a given realm may be limited for a specific set of authenticated clients. A given client can be granted permissions to perform particular operations from the set: subscribe, publish, register, and invoke. The pattern-based subscriptions/registration mechanism allows for the discovery of offered topics and RPCs.

The most recent version of the WAMP specification is from 2021 (164 pages). An initial version of the protocol was released in 2012. The specification defines the basic and the advanced profile. The implemented profile functionality can be discovered dynamically. The advanced profile is considered underspecified and still in evolution. It features trust levels, URI pattern matching, and client listing.

WAMP was created to work over WebSocket, but it can work through any bi-directional and reliable message transport mechanism. One or more WAMP sessions can run sequentially in one WebSocket session. Its specification defines 22 PDUs for the basic profile and four additional PDUs for the advanced profile. The PDUs can carry 14 different message elements. The maximum message length is implementation-dependent; a client can set it between 512 B and 16 MiB. The WAMP protocol defines JSON and MessagePack (https://msgpack.org, accessed on 13 October 2021) notations to serialise subscription/publishing messages and registration/calls of remote procedures. MessagePack gives a more concise representation of messages than JSON. Other serialisation methods may be introduced in future WAMP versions.

There is a considerable collection of programming libraries that include WAMP clients and routers (https://wamp-proto.org/implementations.html, accessed on 13 October 2021). However, many of them support only the basic profile.

Any IoT platform for unconstrained devices can use WAMP. There are some applications of constrained devices that can support WebSocket due to infrequent data transfer. In the case of devices connected to the internet via a pay-per-use plan, careful application programming is needed to minimise communication costs; so, offering remote procedures or subscribing for incoming messages by the devices should be excluded.

Depending on the application’s needs, the functionality of a WAMP router can vary. However, it should be a robust and reliable communication node. An IoT device being a WAMP client, can be used in any communication schema, excluding direct peer-to-peer communication. However, due to WAMP’s symmetric messaging, the devices being WAMP servers can communicate directly.

WAMP can be used in any web application that employs microservices or offers users collaboration, e.g., video games. It also can be used in some IoT systems, i.e., those that can support WebSocket transport. Many IoT devices simultaneously offer procedures that control activators and publish messages from sensors. Moreover, RPCs can serve data querying. The WAMP router can separate such devices from IoT servers, facilitating system integration and increasing its flexibility.

### 4.7. AMQP

AMQP (Advanced Message Queuing Protocol) is maintained by OASIS [12] and standardised by ISO and IEC. An AMQP communication node can act as a producer, consumer, or message queue. A process can contain several such nodes, which enables the building of very complex network components. The process can act as a client, server, broker, or router. The router can forward a message immediately or store it depending on application needs. Connected terminal processes form a peer-to-peer network. The primary communication pattern of the protocol is client-message_router-client. However, an implementation can provide only simple broker functionality in place of the message router.

The AMQP standard defines the encoding of primitive and composite data types, a complex transport mechanism, and a messaging system (i.e., the structure and semantics of messages carried by the transport mechanism). Moreover, the AMQP specification defines how TLS and SASL are used to authenticate and encrypt communications. The encoding allows for binary data representation and also for annotation of data types to support application needs. The transport mechanism enables to open of several parallel communication sessions inside one AMQP connection between two processes. Any number of unidirectional links can be established inside one session. The link serves to transport frames. AMQP can fragment a large message, transmitting it with subsequent frames. The communication end-points advertise the limits of the message and frame sizes. Flow parameters are defined for a link; e.g., message delivery guarantees (QoS) such as the following: at most, at least, and exactly once. Moreover, filtering functions can be associated with the sending node, preventing sending messages that do not meet the specified criteria. In case of a broken connection, the links are recreated, and the state of exchanged messages is preserved.

A message transported by AMQP may have a complex structure. Application data is a mandatory field of the structure. The optional fields are header, transport annotations, message annotations, standard properties, application-defined properties, and footer. Application data and its properties remain the same from the sender to the recipient. The remaining fields can be used and modified by message routers. The standard defines a rich set of elements that can be included in the fields listed above. These elements can influence the processing of messages in intermediary nodes, determine the priority and lifetime of messages, adjust the reliability of their transmission and ensure the required security, and others.

An interesting feature offered by AMQP is the ability to execute transactions, that is, organise groups of messages for joint processing. Transactions can be defined, recalled, and executed. A separate link is created to control the transactions, independent of those for the forwarding of messages.

The rich syntax of AMQP messages and message fields make this protocol universal and flexible. Most of its functions are optional, thanks to which it is possible to tune an implementation size and complexity to the needs of a given application. During the establishment of connections and sessions, the communication parties signalise the functions expected and offered.

The recent AMQP version 1.0 comes from 2012 (125 pages). There is also the WebSocket binding specification [13] from 2016 (18 pages). The initial design of AMQP was carried out at JPMorgan Chase (a financial holding company), and one of its first specifications was published in 2006.

AMQP works on top of TCP. It defines 9 PDUs (called AMQP performatives), which are very complex data structures. It also specifies several dozen data types intended to standardise the structure of exchanged messages. We can distinguish 131 protocol elements defined on the transport, messages, and transaction layers (we counted the number of field name definitions as protocol elements). AMQP can exchange end-user data in a binary, text representation, or any internet media type.

Any IoT platform for unconstrained devices can use AMQP. In the case of devices connected to the internet via a pay-per-use plan, careful application programming is needed to minimise communication costs; the AMQP connections should be as short as possible.

AMQP does not define roles for communicating devices, but it specifies messages that simplify the design of very different and complex application network devices. AMQP can support every communication schema applied in IoT systems. However, in the case of opportunistic peer-to-peer data exchange, it could not be very efficient. AMQP is a good choice for an application that can take advantage of its rich functionality.

The origins of this protocol are related to applications in distributed financial applications. It is mainly used in business applications.

### 4.8. DDS

DDS (Data Distribution Service) is maintained by Object Management Group (OMG) [14]. The standard is open; however, US patents protect some solutions described in this standard. DDS communication services follow the publish-subscribe paradigm, but there is no broker here. Its functions are performed in a distributed way by terminal nodes. In simplified terms, the DDS mechanism is that the publisher writes data to local caches associated with subscribers, and propagation of data between nodes caches occurs automatically. A given node can only be a publisher or subscriber, or both. Defined QoS attributes govern the data transfer process. DDS also automates the switch between the primary and backup nodes when the former fails.

The DDS communication model seen as an API is quite extensive. The publishing application defines thematic channels (named topics). The channel can hold one variable, or it can form a data queue of limited length. In the first case, the subsequent written value (called a sample) erases the previous one. In the second case, the new value is added to the queue. The thematic channel defines the type of transferred values and the set of QoS attributes. A name identifies the channel. Depending on the dynamically set parameters, only one or more publishers can write to a given channel, and only one or more readers can read from it. Moreover, the subscribing application may define filters for the data values received; e.g., it may only receive temperature values exceeding the defined range. QoS attributes are assigned to topics, publishers, and subscribers. An application process defines the attributes, e.g., reliability of the transmission, the maximum frequency of publication, the maximum allowable delay between the received data, or the transmission priority.

DSS allows limiting the range of transferred data by defining domains and partitions within domains. A domain can be viewed as a virtual network. A partition is a collection of communication channels logically related to each other, for example, a collection of sensors in one room, where the domain is a collection of temperature and lighting controls throughout the building. The hierarchy of connections between nodes enables high scalability. DDS provides dynamic discovery of publishers and subscribers within a domain. Although domains are virtually separated, the application can pass data between domains.

The recent DDS version 1.4 comes from 2015 (168 pages). Moreover, OMG published 13 DDS related documents, i.e., extensions, recommendations, and informative statements (1468 pages in total). DDS specifies the object model of the communication system and the abstract API to the objects. Interface Definition Language, also defined by OMG [15], specifies DSS API and data structures. OMG defined the interface between the model and the internet as the DDSI-RTPS protocol [16]. DDSI-RTPS uses UDP and allows for multicast IP addresses. However, DDSI-RTPS does not impose UDP, and other transports, e.g., TCP, are permitted. The modular structure of the protocol allows for a limited implementation on constrained devices. OMG also developed a DDS version (DDS-XRCE) for devices with particularly limited resources [17]. The other DDS related standards define service Invocation (DDS-RPC), Information Modelling (DDS-XTYPES), Security (DDS-SECURITY), as well as programming APIs for C++ and Java. A DDS extension is defined to enable dynamic data type recognition while the system is running.

DDSI-RTPS sends one or more sub-messages in one UDP datagram. Large user data chunks can be fragmented and sent in subsequent messages, and the limit is 216 datagrams (i.e., 4 GiB). Moreover, DDSI-RTPS defines two subprotocols for a participant (a distributed application) and end-point discovery. CDR (Common Data Representation) is the transfer syntax used for DDS data types. The application data can be transferred using CDR, XML, or any application-defined representation. We can characterise the complexity of DDSI-RTPS by two numbers: 12 PDUs (sub-message types) and 94 protocol elements (total number of attributes defined for headers, sub-messages and subprotocols). However, a DDS user interacts with DDS API, which reflexes the DDS objects, not protocol elements. We can characterise the complexity of the API by two numbers: 34 interfaces and 49 structures, which are defined by the IDL specification of DDS.

Secure communication between DSS nodes can be built using IPsec or DTLS. Moreover, the mentioned DDS-SECURITY standard specifies the mechanism of configurable safety plug-ins for DDS software. This standard defines five plug-ins that provide the following services: authentication, access control, cryptographic functions, event logging, and data tagging. Authentication applies to users, i.e., applications that use DDS. Access control allows a given user to grant specific operations rights, such as what DDS domain he can join and what topic channel he can open for publication or subscription. Cryptographic functions include encryption, hash, digital signature, key generation. The event logging plug-in handles all events related to the operation of security services and attempts to breach them. The tagging plug-in allows to label data transmitted through thematic channels to control access rights.

Any IoT platform can use DDS. Moreover, DDS can support every communication schema applied in IoT systems. However, in the case of data querying, it could not be very efficient. Even though DDS allows for direct communication between peers, their discovery and authentication should be assisted by a known server, making such implementation difficult.

DDS is designed for machine-to-machine communication where reliability, performance, real-time operation support, and scalability are essential. Industrial Internet, cyber-physical, and mission-critical are intended applications of DDS.

### 4.9. OPC UA

OPC UA (Open Platform Communication Unified Architecture) is a complex standard for machine-to-machine communication addressed to all industrial domains. The OPC Foundation consortium handles its development. The multi-volume specification of this protocol is available to registered users on the consortium’s website (https://opcfoundation.org/developer-tools/specifications-unified-architecture, accessed on 13 October 2021). OPC UA communication follows the client-server and publish-subscribe paradigms. The second can take a broker-less or broker-based deployment form depending on underlying protocols.

OPC UA is object-oriented. Reach and complex OPC UA Information Model is defined and a schema to represent semantic dependencies between objects. The standard describes the semantics, relationships, and syntax of the architectural elements. Majumder et al. [18] have provided an interesting and detailed comparison of two semantic modelling approaches, the built-in OPC UA and Resource Description Framework (RDF), which is the basis for the semantic web.

OPC UA provides a profile mechanism for the description, classification and discovery of implementation features. The mechanism allows communication between different devices like programmable logic controllers (PLC) and powerful server machines. A common understanding of implemented OPC UA profiles is needed to guarantee interoperation between applications. The profile can be a subject of conformance testing and certification.

OPC UA defines the interactions between servers and clients as well as the set of services offered by servers. An application can act as a server, client, publisher or subscriber or any combination of the four. OPC UA defines address spaces for nodes that represent physical or virtual objects reachable by an application process. The transported messages can be OPC UA-defined or application-defined data types. Servers can provide clients with current and historical data, also alarms and notifications about significant changes. Moreover, servers can store object data model definitions that clients dynamically discover. A client can query servers for the metadata and discover available services and formats for requested data. Moreover, a local discovery server can be deployed to optimise the discovery process.

OPC UA evolved from previous OPC standards based on Microsoft technologies, namely Object Linking & Embedding (OLE), Component Object Model (COM), Distributed Component Object Model (DCOM). After more than three years of specification work and another year for prototype implementations, the first version of the Unified Architecture was released in 2006. Its today’s specification (release 1.04 published at the turn of 2017 and 2018) consists of 19 documents (1683 pages). The first 14 documents (1401 pages) are considered as the main specification. Moreover, two subsequent documents are under preparation. OPC Foundation works with many other organisations to create OPC UA representations of information from different domains. Several dozen documents with UA information models are available on the Foundation web pages. It should be noted that the client-server communication was the base for OPC UA from its very beginning. However, the publish-subscribe communication was added relatively recently (the draft proposal in 2017). We can refer to them as Client/Server and Pub/Subchannels. Due to the youthfulness of Pub/Sub specification, its implementations are still rare.

OPC UA is designed to be flexible and not bound to a particular transport protocol or messaging system. Only specific documentation parts define mappings to selected protocols. Transport of messages between clients and servers can occur directly over TCP, HTTPS, HTTP/SOAP, or WebSocket, using binary, XML, or JSON codings. Transport of messages between publishers and subscribers can run over different messaging middleware, e.g., UDP with IP multicast, MQTT, AMQP. OPC UA defines Pub/Sub communication as an abstract model and mapping to binary and JSON representation. The binary representation is called UA Datagram Protocol (UADP). UADP can even work directly over Ethernet.

OPC UA specifies 25 built-in data types, which include a structure for carrying application-specific data. Moreover, there are 59 standard data types, 17 standard reference types, 32 standard event types, and more. They are used to construct structures, arrays, and messages. A server can implement 40 services. The service requests and responses (with defined parameters) are transported inside a reliable session or sessionless communication via a secure communication channel. We can consider the service requests and responses as PDUs. The number of different defined parameters is 166. Moreover, the OPC UA specification defines an abstract connection protocol that establishes a full-duplex channel between a Client and Server. The protocol consists of 4 PDUs with 15 parameter fields together.

The Pub/Sub communication model defines a complex and hierarchical structure of a network message passed to or received from a communication middleware. Analysing the UADP definition, we can distinguish 3 PDUs and 37 protocol elements. Most of the elements are composite data structures.

OPC UA supports communication with constrained devices and size limits of transport layer frames. OPC UA messages can be sent in chunks. However, an intended security mechanism can impose a weighty value for a message buffer. For example, OPC UA Secure Conversation requires a buffer size that is at least 8 KiB.

OPC UA defines flexible mechanisms with choices to guarantee authentication, authorisation, integrity, and confidentiality. The defined message structure both for client-server and publish-subscribe communication can carry security data. It is possible to use symmetric and asymmetric cryptography, certificates and different deployments of certification servers. Moreover, a diagnostic mechanism is defined that helps to troubleshoot and discover security breaches.

OPC UA is a standard that defines abstract data models, communication scenarios and mapping to underlying transport protocols. Physical performance parameters depend on implementation choices. OPC UA by itself does not restrict any kind of application. It can support any type of device-level IoT platform and any of the seven communication schemas (defined in Section 2). However, not every implementation can be so universal. We can even meet opinions that the support of OPC UA functionalities, communication profiles and services is not practical for IoT devices, e.g., Karaagac et al. [19]. However, the authors did not consider the Pub/Sub implementations.

Burger et al. [20] have compared the OPC UA Client/Server and Pub/Sub communication performance parameters. They made some measurements using selected implementations of the protocol on a Raspberry Pi platform. They found that the server CPU is the main bottleneck for OPC UA Pub/Sub communication and that the Client/Server session management overhead can severely impact performance.

Raddatz et al. [21] give an excellent introduction to the OPC UA Pub/Substandard. Although, their article aims to compare performance parameters of three OPC UA implementation types, i.e., Client/Server, Pub/Sub over UADP, and Pub/Sub over MQTT. They demonstrated that the delay in exchanging a message using the Client/Server communication is relatively high compared to the time required to send the same message using one of the Pub/Sub solutions; moreover, the communication based on MQTT is slower than the UADP variant.

### 4.10. LwM2M

The Lightweight Machine to Machine (LwM2M) protocol was introduced to allow communication and management for constrained devices. The first version of the protocol was developed and published by Open Mobile Alliance (OMA). In late 2017 OMA merged with the IPSO Alliance and released further versions of the protocol as OMA SpecWorks. The most recent LwM2M protocol version is 1.2, which was published in November 2020. The whole specification is freely available at the consortium website (http://openmobilealliance.org/wp/index.html, accessed on 13 October 2021).

The protocol uses a client-server model, in which IoT devices implement client software. The LwM2M servers, because specification introduced one general LwM2M server and one bootstrap server, are places in the public or private data centre and provide services for connected devices. The protocol specification defines four logical interfaces between clients and servers, which allow the following: device bootstrap, device discovery, and registration, device management and service enablement, and information reporting. For each interface a unique list of operations could be used, and their “direction” is defined. For example, in the device management and service enablement interface, only the server could perform some operations on the client, such as reading, writing, executing, creating, deleting, and discovering. In contrast, the information reporting interface server could perform four operations on the client (observe, observe-composite, cancel observation, and cancel observation-composite). The client could perform two operations on a server—notify and send. The defined operations are mapped to the sentences of a RESTful interface, and they can be perceived as PDU types; their number is 26. The corresponding URI path types and query parameters can be perceived as the protocol elements; their number is also 26.

LwM2M introduces not only a communication protocol but also a data model. The data owned and/or managed by the IoT device is provided to the server as a resource. Resources are logically grouped into an object. Depending on the specification and the resource type, the server could read, write and execute the given resource. Each IoT device has to provide an object called “device”. This object contains over a dozen resources that could be read, for example, manufacturer, model type, serial number, and battery level. Some resources could also be written, for example, current time and time zone. Some could even be executed, for example, reboot, factory reset, or reset error code. Some objects are mandatory and have to be provided by each IoT device that uses the LwM2M protocol. For example, the LwM2M server and LwM2M security belong to such a class of objects. OMA 3rd party standard development organisations and individual vendors registered many additional objects. We could find elementary objects like digital input, analogue input, or more complicated sensors, for example, smoke alarms and CO detectors. A complete list of registered objects is provided at the consortium website (https://technical.openmobilealliance.org/registries.html, accessed on 13 October 2021).

As even protocol name highlights, it was developed for constrained devices. The first version of the protocol, 1.0, uses a limited subset of the CoAP functionality and allows transmission of LwM2M messages using CoAP/UDP or CoAP/SMS. Both communication channels could use DTLS for security. In the most recent 1.2 version of the protocol, new carrier protocols are introduced, HTTP and MQTT, carrying data formats chosen by a developer. In the most recent 1.2 version of the LwM2M protocol, data could be encoded using, among others, plain text, octet strings, TLV, CBOR (Concise Binary Object Representation), and JSON. The only payload limit size concerns TLV encoding, which is mandatory for the LwM2M 1.0 version and is set to 16.7 MiB for any given resource and object.

The first version of the protocols was devoted to the constrained devices. During further revisions of the LwM2M protocol, other transport protocols are introduced. With its advantages and disadvantages, selecting a given transport protocol impacts the whole LwM2M mode of action and could completely change its behaviour and resource demands.

### 4.11. Proprietary Protocols for Smart-Home Devices

Smart-home systems constitute a significant segment of the IoT market. Their design can be based on the presented above messaging protocols. However, three proprietary protocols, i.e., Google Weave, Amazon Alexa Mobile Accessory Kit Protocol, and Apple HomeKit Accessory Protocol, have gained popularity due to corresponding applications available for smartphones and home computers. Thanks to the applications, manufacturers of domestic appliances can swiftly find customers. The application users can remotely manage these devices from a special control panel or a computer, tablet, or smartphone equipped with one of those applications. In most cases, these systems utilise wireless solutions, such as Wi-Fi, Bluetooth low energy, or Zigbee for connectivity. However, the IP protocol usage enables remote managing of smart devices from any point of the globe.

One of the reasons for the popularity of the three smart-home applications is their associated voice assistant: Amazon Alexa, Apple Siri, and Google Assistant, respectively. The three companies offer voice assistant devices, i.e., Amazon Echo, Apple HomePod, and Google Nest Mini. Moreover, they maintain cloud environments that allow remote access to home devices worldwide. In consequence, they control their smart-home systems enabling the design of domestic appliances to other enterprises. Unfortunately, only the Google Weave protocol specification is publicly available. The details of the other protocols are available after signing a business agreement. Due to this constraint, we describe them rather generally. Further in the comparison, only Weave is considered and compared with the other messaging protocols.

#### 4.11.1. Google Weave

The Nest Labs initially developed the Weave protocol in 2013. It allows communication and cooperation between various smart home devices, for example, thermostats, security cameras, or smoke detectors. The protocol was designed for the so-called Home Area Network (HAN) and enabled communication between HAN and the Nest cloud. In effect, smart-home devices could be managed or monitored remotely via mobile or the internet. After acquiring Nest Labs, Google provided its own reference implementation as an open-source library named OpenWeave. All information concerning the Google software library and documents concerning the Weave protocol are provided on the project website: https://openweave.io (accessed on 13 October 2021). The first version of the Weave specification is from 2013, and the most recent ones or new revisions are from the beginning of 2020.

Weave uses the publish-subscribe model. Each smart device contains one or more traits, which represent its possible state by various properties. For example, the audio trait has properties that describe volume value and mute state. Users or other devices can remotely change these properties, and other devices or services that subscribe to a particular property will be notified. Traits could be joined together to describe more complicated functionalities in so-called interfaces; for example, the intercom interface can consist of microphone and speaker traits. Several traits and interfaces form so-called a resource, which in most cases represents a device. In the Weave HAN, each resource is identified by IPv6 address with the fd00::/48 prefix. Resources could be associated with a physical device, or they can form a kind of virtual resource, which is called a controller. The controller using its higher processing power could add more complicated functionalities to the device traits. One kind of controller is the resource proxy controller, which can be used for routing messages and providing state information of the device. For example, it can switch the device off periodically to save battery.

Weave protocol, in reality, consists of multiple protocols (called in documentation as Profiles), for example, Echo, Heartbeat, Bulk Data Transfer, and essential Weave Data Management (WDM). WDM allows requesting current traits’ property values, changing them, and providing notifications. For this purpose, five basic operations are used: subscribe, notify, observe, update, and command. The first one allows subscription to the interesting trait property. After that, all changes of the property cause sending of notification for the subscriber. Observe allows querying of current property state. The update operation is used to change the property values, which allows managing the device state. The last operation, command, enables the execution of specific functions provided by the trait and could not be changed by a property update. For example, this operation could be used for complicated tasks like choosing the next track on a media player or starting a neighbouring Wi-Fi network scanning process.

Weave messages due to performance reasons and support for constrained devices use the TLV communication scheme. They are transmitted via the UDP or TCP protocol carried by IPv4 or IPv6. Moreover, they can be directly included in the BLE or Thread frames for low-range communication in the HAN.

The open-source OpenWeave library, provided by Google, utilises an approach similar to the RPC implementations. First, the programmer defines all necessary traits, interfaces, and resources using a special language called Weave Schema Description Language (WDL). After that, a dedicated compiler generates source code, which could be compiled in the target environment after necessary enhancements. For the embedded and mobile devices, this tool will generate C++ with Weave TLV code. For services running in the cloud and mobile apps Objective C, Swift, Java, and Scala could be chosen.

#### 4.11.2. Amazon Alexa Mobile Accessory Kit Protocol

The simplest way to connect an appliance to the Alexa voice assistant is to use the Z-Wave or Zigbee radio stack. In this case, controlling the appliance is performed using features introduced in the application layers of these two radio stacks. Both stacks define profiles for most common device types, for example, thermostats, light bulbs, switches, or fans. For each category, well-defined voice commands could be given to the Alexa assistant, and it manages the appropriate device. For example, “Alexa, turn on the lights”, “Alexa, raise the temperature 1 degree”, or “Alexa, ask Garageio to close my garage”.

The second option for connecting an appliance to the Alexa system is dedicated to devices with built-in microphones. In this solution, a developer could use Alexa Voice Service (AVS). Depending on the appliance’s computing resources, this solution can use voice recognition of so-called “Wake Word” or a simpler method called “Tap-to-Talk”. When the command issued will be detected in both situations, the appliance records a short voice sample, which is later sent to the AVS for analysis. For this purpose, the HTTP/2 protocol is used as a transport protocol. All data, including voice samples, are encoded using JSON. As an HTTP/2 response containing a JSON structure with directives to the device or exception—in case any errors are received.

The last method for connecting an appliance to Alexa Cloud utilises Alexa Connect Kit (ACK). ACK is a dedicated hardware module produced by the Espressif company. This tiny electronic board (16 × 23 × 2.3 mm) contains Wi-Fi and Bluetooth Low Energy capabilities and runs Amazon custom firmware. This firmware carries out all communication mechanisms associated with connection to the Alexa Cloud. A newly developed appliance exchanges messages with the module using the UART (Universal Asynchronous Receiver-Transmitter) interface. Amazon provides a dedicated SDK (Software Development Kit) for the device microcontroller, seamlessly hiding all UART communication details and providing intuitive interfaces for the most popular devices. The developer has to only react in the appliance hardware to appropriate commands, e.g., switch on the light. Currently, this setup allows managing fourteen interfaces for the most popular devices, such as cooking appliances, power controllers, or temperature sensors.

#### 4.11.3. Apple HomeKit Accessory Protocol

HomeKit is Apple’s proprietary protocol for smart-home devices. The devices could be commanded and managed by the Apple Siri voice assistant or Home app, which works on Apple products, like iPads, iPhones, and Apple Watches. The workhorse of this solution is HomeKit Accessory Protocol (HAP). It uses two radio mediums—Wi-Fi with IP and Bluetooth Low Energy (BLE), according to the residual publicity available documents. HAP introduces around thirty device categories of popular device classes, such as air conditioners, IP cameras, smoke alarms, or thermostats.

#### 4.11.4. Matter Protocol Formerly Connected Home IP

The existence of three not-compatible smart-home ecosystems leads to inconveniences for clients and device manufacturers. They must choose one ecosystem and buy or develop devices compatible with the chosen one. Due to this fact, Amazon, Apple, Google, and Connectivity Standards Alliance (formerly Zigbee Alliance) decided to ally and develop one common standard for smart-home devices. From December 2019, when the project was launched, the consortium named the project Connected Home IP, shortly mentioned as CHIP. In May 2021, the name of the project was changed to Matter. All further details should be provided on the project web page (https://buildwithmatter.com, accessed on 13 October 2021).

We could learn, from available information, that their messaging protocols extensively will be applying the IPv6 network protocol plus TCP and UDP as the transport protocols. It will use various communication protocols in the lower layers, including Ethernet, Wi-Fi, Bluetooth Low Energy, and Thread, utilising 802.15.4. The most crucial design decision concerns the usage of the open-source approach. The first specification of Matter should be publicly available in the first quarter of 2021, according to the earlier press announcements. However, at the time of writing, no detailed description is available.

## 5. Comparative Analysis

In the previous chapter, we characterise the most popular protocols used in IoT systems. In this chapter, we present a comparison of them. We decided to use a qualitative rather than quantitative approach. The rationale for this decision is that presenting one representative situation is almost impossible due to the complexity of the protocols, the number of theirs extensions, and possible usage scenarios. Moreover, all quantitative measurements in such an environment would be valuable to a very narrow group of readers with an almost identical system. We are convinced that our qualitative analysis will be beneficial for a broader group of designers. Additionally, comparison, presented in the following text, allows a better selection of protocol for a given system design.

We structured the analysis into three parts. The first sees to selected functional features of the messaging protocols, the second deals with their maturity and complexity. The third considers their applicability to the defined (in Section 2) types of device-level IoT platforms and suitability for different communication schemas.

### 5.1. Functionalities

The formation of any protocol was led by an intended purpose and related features. Table 1 summarises them. If a system under design is in line with the primary purpose of a protocol (see the second column), then the protocol will simplify the design. As we see, MQTT-SN and CoAP were conceived for communication with constrained devices and MQTT for the end devices that have limited memory and link bandwidth. As time goes by, protocols evolve, and their extensions appear, e.g., the DDS-XRCE extension, which supports constrained devices and their access to the standard DDS cloud via a gateway.

The protocol specifications define specific roles for the communication points and the target architecture of the end-system. The two features are good indices for the selection of a protocol. However, with some programming effort, the application can adapt to other roles and architecture. Only WAMP defines RPC related roles, but a programmer can use any other protocol that allows two-way message exchange to achieve RPC calls. An application can also take profit from the discovery mechanism provided by the selected protocol. If there is no such support, the programmer should craft some code to achieve what is desired.

The real-time support cannot be added on the top of a protocol if it does not have it. As we see, CoAP, DDS, OPC UA, and LwM2M provide that feature. However, not every implementation of them guarantees real-time backing—e.g., the selected underlying transport protocol can limit this feature. The partial support means that the transmission time can be estimated. However, the broker/router/server delays are difficult to be predicted, as the devices are prone to congestion. Thus, only soft real-time applications can use such protocols as MQTT, STOMP, and AMPQ. In WAMP, an application node can play any role or roles, so we can build communication between two nodes without an intermediary, which can be considered real-time support. However, the typical WAMP use-cases are with broker or dealer, then we consider real-time support partial. Some applications, e.g., instant messaging, expect real-time behaviour only during a communication session. This kind of application is insensitive to delays related to opening the session, and the full speed TCP connectivity is considered sufficient real-time condition.

The specific features given in the last column strongly distinguish the protocols. If they are required, the protocol selection is straightforward.

Table 2 presents some selected features related to application messages to be exchanged. In IoT systems, the messages are usually short, and the maximum payload size is not an issue. The only exception can be the need for software updates. However, the limit of 64 KiB for updates is not restrictive for constrained devices, but the more powerful devices enable storing bigger chunks of code. The constraint on energy consumption is a stronger limit for transmitted data volume. The designer of an update subsystem should analyse restrictions imposed by used operating systems and communication middleware. The protocols do not impose prohibitive restrictions; their implementations set such limits.

All of the protocols define transfer representation for their header fields and other protocol elements. Some define allowed representation for application data, carried as the protocol payload; e.g,. XMPP can carry only text representation, and WAMP allows for JSON text and binary MessagePack representations. Only XMPP does not allow for the transmission of compressed data. All other protocols allow carrying binary data, which can be a compressed form of any information. MQTT transport labels in text form, while MQTT-SN supports their compressed representation. If a protocol transmits an unspecified byte stream, then the representation is application-specific. The internet media types (used by CoAP, STOMP, AMQP) specify the data representation explicitly. DDS uses only CDR coding, and WAMP states the MessagePack coding explicitly in the message header. The defined payload representation helps to connect independently designed subsystems.

The labels and metadata simplify interoperability. The labels help distinguish the semantics of exchanged data. The hierarchical structure of the labels, either MQTT-like or URI-like, simplifies querying data sets or filtering them. The metadata can be defined by a protocol, application, or both, to carry additional information about the messages, their transport, and sender or receiver expectations. Some metadata can control the behaviour of message caches, brokers, routes or servers. The protocol labelling and metadata support help the interoperability of complex systems that are programmed by different teams.

Some applications should perform atomic operations on a data set that is formed by several messages. After the messages transfer, the operation can be committed or discarded. Protocol support for transactions is desired when several remote processes should simultaneously execute an atomic action. Only AMQP, DDS, and OPC UA provide such support.

The choice of a transport protocol influences the features of an application protocol. Any messaging based on UDP is more efficient on wireless links with high packet loss rates than protocols based on TCP, which results from the TCP congestion avoidance algorithms and slow TCP connection start mechanism. The efficiency gain is observed as lower latency. UDP delivers faster short messages than TCP without losing time on connection setting up. Moreover, UDP-based traffic has lower transmission overhead due to the small UDP header size and absence of overmuch acknowledgement packets. The reliability of TCP is an advantage. It is helpful to transfer big chunks of data, freeing the programmer from dealing with the network congestion control. However, TCP acknowledges data delivery, not processing, which should be approved by the application if needed. Such approval is a must in machine-to-machine communication or if the TCP connection is prone to losses. The application layer acknowledgements make the transport ones redundant. Moreover, standard TCP implementations are memory demanding, which is prohibitive for constrained devices. RFC 9006 guides TCP usage in IoT systems. There are even some TCP/IP stack implementations for constrained devices (e.g., uIP, IwIP, GNRC/RIOT) with reduced functionality and efficiency. Their purpose is to enable communication with TCP based servers and applications when it is not to avoid. Running application protocols on the top of UDP is a better choice for constrained devices or high loss radio networks, which is confirmed by many experiments devoted to performance evaluation of messaging protocols, e.g., Moraes et al. [22] and Thangavel et al. [23].

Table 3 shows the primary transport protocols defined for the analysed protocols. The additional ones are given according to the IANA registrations. As we can see, MQTT-SN, CoAP, and DDS are the only protocols running by definition over UDP. An OPC UA application can use at the same time, both TCP and UDP or only one of them. WebSocket popularity goes from the need of overcoming communication problems related to restrictive firewalls, which exist in enterprise and institutional networks. It allows for opening a TCP connection using the standard HTTP port, which is seldom blocked. WebSocket exchanges two HTTP messages on opening and allows multiple data streams (text or binary) over one TCP connection. Moreover, some extension data can be sent together with payload data. The extension data may carry metadata information. These features can be used by an application protocol and be specified as a so-called WebSocket profile or subprotocol. Some messaging protocols have such profiles defined.

Transport Layer Security (TLS) and Datagram Transport Layer Security (DTLS) protocols aim primarily to provide privacy and data integrity between communicating processes. They also allow authenticating one or two communication end-points. TLS works over TCP. DTLS is its functional equivalent that works over UDP. These protocols use public-key cryptography for secure key exchange. When the encryption key is securely exchanged, user data for efficiency purposes are encrypted using symmetric encryption, in most cases some variant of AES encryption. Moreover, data stored in the certificates could be used for the authentication of one side of communication. In most cases, in such a way server where clients are connecting is verified. However, TLS/DTLS introduces so-called mutual authentication during both communication sides are authenticated. In the IoT environment, this feature can be used for authentication of IoT devices as well as a central server, where they send data. TLS/DTLS can also be configured to work with pre-shared symmetric keys (RFC 4279). Thus, an IoT system designer can based authentication on digital certificates with public keys, raw asymmetric keys, and pre-shared symmetric keys.

Another alternative for providing security and authentication is using Simple Authentication and Security Layer (SASL) specified in RFC 4422. This protocol, called by authors, framework, allows the addition of security functions for currently used application protocols, like that used in the IoT environment. In contrast to TLS/DTLS, it contains an extensive list of possible authorisation mechanisms, such as simple plaintext, various challenge-response mechanisms, and integration with Kerberos or OAuth. It could introduce data encryption and integrity checking; however, it allows the usage of TLS for this purpose. XMPP and AMQP could utilise both mechanisms, TLS and SASL, at once.

StartTLS mechanism (a.k.a. Opportunistic Encryption) defined in RFC 4322 allows starting an encrypted connection without any pre-arrangement specific to the pair of systems involved. StartTLS is used to initiate an encrypted connection on the same port as an unencrypted connection.

Every TCP session can be secured by TLS and every UDP session by DTLS. Both TLS and DTLS mechanisms have many options, so application profiles define the particularities. Some messaging protocols have such profiles. Moreover, as column 3 depicts, some of the protocols specify additional mechanisms for authorisation purposes.

The protocols differ in the way they handle QoS parameters. XMPP, AMQP, and DDS support priority data transfers. A messaging protocol can support the reliability of data delivery. The support is needed if unreliable transport is used (UDP) or intermediate devices (brokers, message routers, or servers) are on the path. The expected reliability level is expressed as the QoS parameter in MQTT, CoAP, and AMQP. However, DDS offers the most extensive set of QoS settings.

Application messages can be identified differently. Which one is the most suitable depends on the application logic. Column 6 shows the offering of the analysed protocols. Querying and subscriptions can be more efficient if the addressing is supported by a filtering capability (column 7).

### 5.2. Maturity and Complexity

All of the messaging protocols can be found in many existing deployments. We assess their maturity seeing the time its stable specification exists. From that perspective, only WAM can be considered as not mature. Figure 1 depicts the time of the first draft of the protocols appeared, and older specifications were valid (the yellow colour bars), the years of the latest specifications (the beginnings of grey bars), and the publication years of the newest protocol extensions (the starts of green bars). We can see that simpler protocols tend to be more stable, as not many modifications or extensions are proposed to them.

The complexity of a protocol determines the time and cost of learning. Moreover, very complex protocols can be challenging to use, thus, entail more programming errors or suboptimal coding. We can estimate the complexity of the protocols by comparing the volume of their specifications (Figure 2) and the number of defined protocol elements (Figure 3). We distinguish volumes of the core protocol and related standards. The comparison is not very strict. In most cases, the related standards propose some functional extensions, but they can give implementation guidance, use-cases descriptions, or user APIs. For example, the DDS related documents cover many API issues, while MQTT none.

For LwM2M, we only provide the volume of the core specification. It is difficult to assess the volume of related documents, as they are XML schemas for object modelling and registries of objects and resources. The same for OPC UA—we have not calculated the volume of many information models already published. Nevertheless, the effort needed to learn the full potential of DDS, XMPP, OPC UA, LwM2M, and CoAP is more significant than to learn STOMP, MQTT, or WAMP. We should notice here that there are more information models for IoT, which are defined by other organisations, e.g., W3C “Web of Things (WoT) Thing Description”.

The complexity of the protocols is illustrated by the number of PDUs and protocol elements. The PDU number reflects how many different operations an application can call. The number of elements reflects how many parameters a programmer can set. For the Weave protocol, PDUs reflect five possible actions that could be performed on so-called traits, which describe each device’s functionality. These traits could be in this comparison treated as protocol elements. We cannot show this number, as it depends on the device’s functionality. Moreover, the bigger protocol complexity, the bigger footprint implementation has. However, the footprint size can depend on the application code and implemented options. The middleware and code development environments, especially for constrained applications, optimise the size of linked libraries according to their usage. From the complexity measures perspective, CoAP is simpler than MQTT, contrary to documentation size. The first perspective indicates the learning time the second programming difficulty.

The number of PDU types determines the number of SDUs (Service Data Units), which are the API functions a programmer interacts with. The API of a given protocol library is slightly more numerous in practice than the PDU type number. The API provides functions that instantiate and close the protocol context. Moreover, it optionally provides functions that alter the context state, process some data format conversions or provide some security operations. With a given messaging protocol implementation, the available programming libraries often provide rich functionality, e.g., a complex server or broker module. The programmer should pay attention to the protocol version and options supported by the selected library. The library may implement one of the first protocol versions, and that the library may provide some extensions not yet standardised. Moreover, two libraries can support different sets of protocol options, which results in interoperability problems.

### 5.3. Suitability and Applicability

We have distinguished four types of IoT devices related to communication needs. Table 4 shows the suitability of the analysed protocols for them. The constrained devices that save energy should minimise the volume and frequency of transmitted data. The constrained devices with a small memory size and processing power need simple protocols with a small footprint. The two requirements imply the choice of UDP-based protocols (MQTT-SN, DDS, UADP). Some TCP-based protocols (MQTT, STOMP, LwM2M) can be selected if the expected data transfer is infrequent and short.

The devices whose cost of internet access should be considered (e.g., those connected via a low-power wide-area network) should minimise the time of communication sessions and volume of transferred data. Hence, the protocols based on long TCP connectivity should be avoided (XMPP, WAMP, AMQP). All the analysed protocols can tolerate session interruptions and are suitable for always-online devices.

We have distinguished seven communication purposes for IoT applications. As we see in Table 5, most of the purposes can be realised using any protocol. The only exception is the opportunistic peer-to-peer data exchange. For example, such exchange is necessary for communication between mobile devices. Direct peer-to-peer communication needs symmetric roles for message exchange and a lack of intermediate devices. Only CoAP can easily be used for this purpose. DDS allows for direct communication between peers; however, their discovery and authentication are a problem to overcome in a way. WAMP specification promises communication with symmetric roles between neighbouring devices. Nevertheless, it could be more suitable to create links between WAMP brokers than between mobile devices.

The IoT applications that expect data samples only on request can use any of the protocols. However, using some of them may be suboptimal, as column four shows. For example, MQTT and MQTT-SN cannot be optimal for process control applications. The intermediary devices introduce delays. Moreover, the intermediaries can be overloaded by such applications.

Information concerning Weave can be a little misleading. Weave can be used, or at least have functionality, for all considered communication purposes. However, due to its specific application, in most cases, this protocol is used only for smart-home device management and utilised only by dedicated smart-home management software or smart-home cloud.

## 6. Related Work

We can find many papers that compare selected messaging protocols. However, most of them analyse only a few of them. Moreover, the older comparisons are outdated due to the continuous evolution of the protocols, as Figure 1 depicts. Below we give a short review of the recent papers trying to supplement our comparison with interesting information and illustrate the reading available on this topic.

The need to find an appropriate messaging protocol for a specific application has stimulated some research. For example, Amaran et al. [24] have evaluated CoAP and MQTT-SN in a robotic application. Their experiment shows that MQTT-SN performs 30% faster than CoAP. Durante et al. [25] analysed the same two protocols for a marine environment acoustic monitoring system design. They demonstrated that MQTT-SN latency is 30% lower, the power consumption is 10% lower, and the traffic flow is 2.15 times larger than CoAP for architecture with 40 wireless nodes. The two works selected only MQTT-SN and CoAP, as both protocols work over UDP, which is a justified decision for constrained devices.

Smart grid systems are another specific IoT application, which motivated the Šikić et al. work [26]. Their laboratory experiments showed that the MQTT protocol achieves minimal message overhead and shortest delivery time than AMQP and HTTP. Glaroudis et al. [27] analysed a more extensive set of protocols for smart farming developments. They have compared MQTT, CoAP, XMPP, AMQP, DDS, REST-HTTP, and WebSocket. They conclude that the most promising protocols for agriculture applications are CoAP (when regarding such factors as latency over LAN, bandwidth consumption, and energy consumption) and MQTT (considering latency over a mobile network, throughput, reliability, developers’ and researchers’ preferences). They admit that there is no suitable-for-all solution, and different protocols can be reasonable for device-to-gateway, gateway-to-cloud, and cloud to the end-user communication.

Dizdarević et al. [28] have compared a similar set of protocols, i.e., RESTful HTTP, MQTT, CoAP, AMQP, DDS, and XMPP, focusing on possible implementations in the IoT-based fog and cloud computing systems. They conclude that the most mature are RESTful HTTP and MQTT, that MQTT has excellent performance on constrained devices, and that the performance of RESTful HTTP is not sufficient for combine IoT-fog-cloud solutions. They notice that AMQP has relatively high power-, processing- and memory-related requirements, making it a rather heavy protocol, not well-fitting IoT systems. They mention that XMPP has some inconveniences, i.e., massage size, absence of QoS, lacks an efficient binary encoding, and that the lightweight XMPP publish/subscribe scheme is not yet available. They have analysed several dozen papers presented comparisons to assess latency, bandwidth utilisation and throughput, energy consumption, and security of selected messaging protocols. The observation is that the optimal protocol choice depends on the selected application scenario; e.g., it depends on the size of the payloads and the transfer’s frequency and burstiness. Consequently, they conclude that the straightforward solution would include combining a lightweight protocol between IoT and the fog and a protocol not restricted to the constrained devices between the fog and the cloud. They consider two protocol pairs CoAP with RESTful HTTP and MQTT with AMQP.

Ghotbou and Khansari [29] have identified a set of particular requirements that should be satisfied for video transmission in low-power lossy networks. They analysed the suitability of many protocols for that purpose, i.e., AMQP, CoAP, DDS, MQTT, MQTT-SN, Websocket, XMPP, HTTP 1.1/2.0, RTP/RTCP. They concluded that CoAP is the most suitable protocol amongst all reviewed, and CoAP could satisfactorily support video transmission on constrained and non-constrained networks.

One of the aims of vehicle-to-cloud communication is maintaining digital twins. Proos and Carlsson [30] have compared the performance parameters of AMQP, CoAP, MQTT for such applications. They used real-case data and selected protocol implementations in their experiments. Their results show that CoAP has the lowest latency and overhead but cannot guarantee reliable transfer (even when using its confirmable message feature). The best performer that guarantees reliable transfer is MQTT. We can comment that TCP, similarly to CoAP, has a retransmission counter, which breaks the connection when it expires. Thus, their observation shows a flaw in the implementation, not on the protocol side.

Talaminos et al. [31] have analysed the DDS, MQTT, CoAP, JMS, AMQP, and XMPP protocols considering a particular e-health use-case, i.e., monitoring respiratory rehabilitation of chronic obstructive pulmonary disease patients in ambulatory and at home. They built a dedicated benchmark framework and gathered different performance metrics, including CPU usage, memory usage, bandwidth consumption, latency, and jitter. They conclude that DDS is the best choice for the ambulatory scenario and MQTT for the home scenario. Their metrics well illustrate the strengths and weaknesses of the analysed protocols.

The selection of appropriate messaging protocol for an industrial application is a problem analysed in some recent papers. Karaagac et al. [19] have analysed the OPC UA and LwM2M protocols and implemented a docker-based virtualisation server to enable cooperation between networks built over the two protocols. They pointed out that OPC UA supports more complex data models (e.g., type inheritance, nested object structure) than LwM2M. Moreover, they notice that OPC UA supports a wide variety of data types and methods, where LwM2M only defines eight data types and uses existing CoAP methods. Profanter et al. [32] have compared performance parameters of selected implementations of OPC UA, ROS (Robot Operating System), DDS, and MQTT. They demonstrated that the OPC UA and DDS implementations deliver high performance than the ROS and MQTT implementations.

In the research by Thangavel et al. [23], we can find a performance evaluation of MQTT and CoAP based on a common middleware and application scenario. Their findings reveal that MQTT messages have lower delay than CoAP messages at lower packet loss rates in a transmission medium. However, the results are opposite at higher loss rates. Moreover, when the message size is small and the loss rate is equal to or less than 25%, CoAP generates lower additional traffic than MQTT to ensure message reliability. We can notice that their findings stem from the features of TCP and UDP that carry PDUs of MQTT and CoAP, respectively.

V. Sarafov [33] has constructed an abstract theoretical model for comparing the overhead of the same protocols, then the article above-mentioned presenting experimental comparison, i.e., CoAP and MQTT. Moreover, Sarafov also analysed WebSocket. He confirmed the theoretical results with some experiments. He showed that the overhead of CoAP with non-confirmable requests and responses is the least. Next, MQTT with QoS 0 is the second. However, CoAP’s reliable configuration performs better than MQTT QoS 2. WebSocket behaviour is slightly worse than MQTT due to the additional data exchange for connection opening.

Recently Al-Masri et al. have published a comprehensive comparison of messaging protocols for IoT [34], i.e., HTTP, MQTT, CoAP, AMQP, DDS, and XMPP. They analysed the support of the protocols by ten popular IoT platforms (like Azure IoT Hub, Google IoT Core). Interestingly, all platforms carry HTTP and MQTT; six support AMQP, four CoAP, two XMPP, none of them DDS. The authors analysed about 170 papers to gather helpful information about the protocols, providing long lists of the protocols’ advantages and disadvantages. Even though some of their statements are arguable, we consider their study a valuable review of numerous analyses of the messaging protocols. We find their findings complementary to our comparison presented in the previous chapter.

Another recent comparison of IoT messaging protocols by Silva et al. [35] provides experimental data on MQTT, CoAP, OPC UA usage. The article also analyses several communication techniques in the context of IoT design, namely HTTP, CoAP, QUICK, AMQP, MQTT, DDS, OPC UA, and NDN (Named Data Networking).

In the paper by N. Naik [36], a short comparison of MQTT, CoAP, AMQP, and HTTP can be found. He ranks the protocols concerning message overhead, resource consumption, bandwidth, latency, reliability, security, and usage. The ranking is well presented. However, the reliability and security features depend more on implementations than on the protocols themselves; thus, these features analysis is arguable. Unfortunately, we have found more articles presenting arguable methodology or conclusions.

Many of the related works investigated the performance parameters of selected messaging protocols. The consumption of computation resources strongly depends on the implementation way, installed and activated extensions, configuration method, and in some cases also the way the application is used—for example, a protocol throughput depends on the size of data samples, on the QoS setting. In consequence, the performance comparisons should be cautiously considered. Moreover, functional comparisons from the past may no longer be valid due to the evolution of the standards and implementation libraries.

We tried to gather the performance measurements from different articles and put them together, but such data was misleading. The published results are incomparable. They are related to different usage scenarios and software and hardware platforms applied in the performed tests. Additionally, presented in the article, comparisons cover only selected protocols, in many cases only two. Due to this fact, we decided qualitatively compare all protocols.

## 7. Conclusions

We have surveyed the plethora of messaging protocols available to IoT system designers and comprehensively compared them. We have also analysed the functional objectives of protocols, their messaging and transport features, complexity, and suitability for different uses. The collected observations and recommendations provided a pragmatic view helpful for IoT system architects. The value of such a view results from the fact that selecting an appropriate communication protocol could have a crucial impact on cost, time, and the most crucial issue—the success of the deployment.

Each of the described protocols has a large group of enthusiasts who promote and develop their standards. All of the protocols are used in many deployed systems. Moreover, each of them is associated with numerous programming libraries. However, some of these libraries are older and do not fulfil all the functions defined in the latest standard versions. There are also functions implemented in a given library but not defined in the standard. All the facts increase the difficulty of choosing the appropriate protocol for a given IoT solution.

The communities engaged in particular protocol evolution tend to make the protocol universal and commonly used. Thus, the results of suitability and applicability analyses (Table 4 and Table 5) do not allow for easy protocol classification. Consequently, the choice of a suitable protocol for a given application is more complicated.

For resource-constrained IoT devices, MQTT and CoAP are attractive protocols. Devices that do not support the TCP protocol due to energy consumption can only use CoAP, MQTT-SN, DDS, and UADP. The significant advantages of CoAP and MQTT over other protocols include low header overhead, low consumption of computing resources, and low message delivery delays. All this makes them attractive for any IoT device. In turn, striving to minimise the complexity of the message exchange mechanism, which translates into computing requirements for the hardware and design time, it is worth considering the mentioned ZeroMQ and YAMI4 communication libraries.

While designing simple systems without particular implementation constraints, the most straightforward protocols, such as STOMP or WebSocket, may be the optimal choice. On the other hand, when we expect functionally rich support in managing messages, LwM2M, AMQP, DDS, and OPC UA, can be attractive. The DDS and OPC UA are the most complex of the protocols presented here, and they can be recommended as a basis for large IoT systems.

A different choice may be optimal for each specific implementation. The protocols of moderate complexity are XMPP and WAMP. It is a good design practice to select a protocol that is as simple as possible that meets the functions necessary for a given implementation. The more complex the mechanism, the more time it takes to design and implement, the more compute and memory resources are used.

The presented comparison is more qualitative than quantitative. It is not easy to find qualitative parameters for comparing protocols basing on their specifications. Even simple parameters like min/average/max header and message sizes depend on application scenarios and selected protocol options, and we failed to calculate them. The interesting parameters like processor, memory, bandwidth consumptions, effective-to-transmitted bitrate are implementation-dependent. To make a credible protocols comparison based on their implementations, a team should build at least three different IoT communication scenarios based on at least three different implementations of each protocol on the same hardware devices. Moreover, even if such a comparison was provided, there was a high risk that results could be completely different in a slightly changed application or due to the usage of other extensions. It is probably a challenging future work to do, but it is huge work. It is also challenging to define some benchmarks for such comparison.

In this paper, we have considered mainly the communication aspects of the messaging protocols. The data representation issues, we touched only in the context of message compactness. The protocols use different approaches to object modelling, expressing their semantic, defining namespaces. An IoT system designer has multiple choices. Thus, there is a need for a comprehensive analysis and comparison of IoT data and metadata representation, which points to an essential direction for further research.

## Figures and Tables

**Figure 1 sensors-21-06904-f001:**
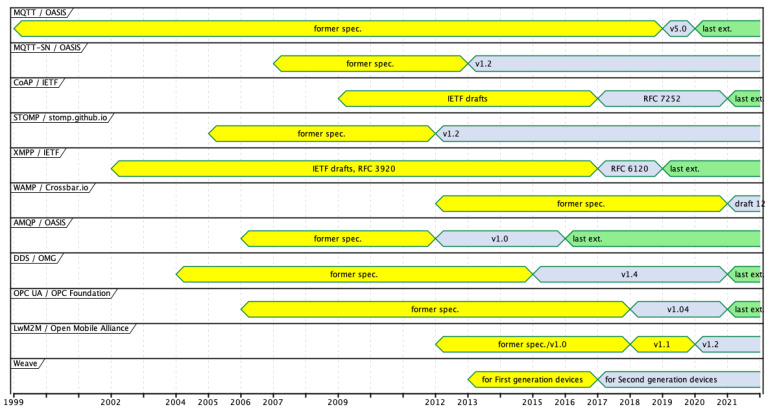
The years of the protocol specification appearance.

**Figure 2 sensors-21-06904-f002:**
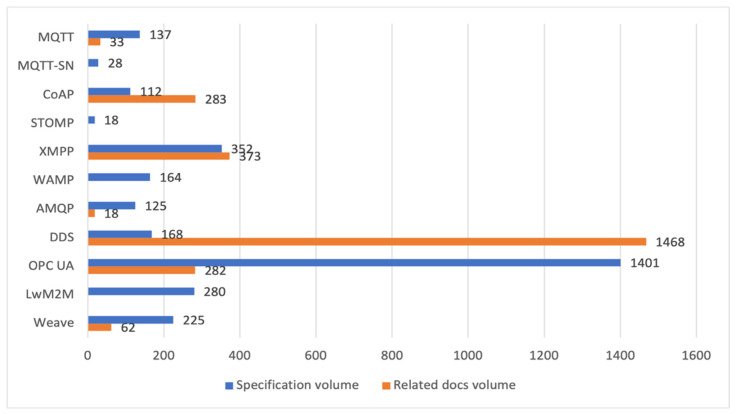
Volume of the protocol documentation in the number of pages.

**Figure 3 sensors-21-06904-f003:**
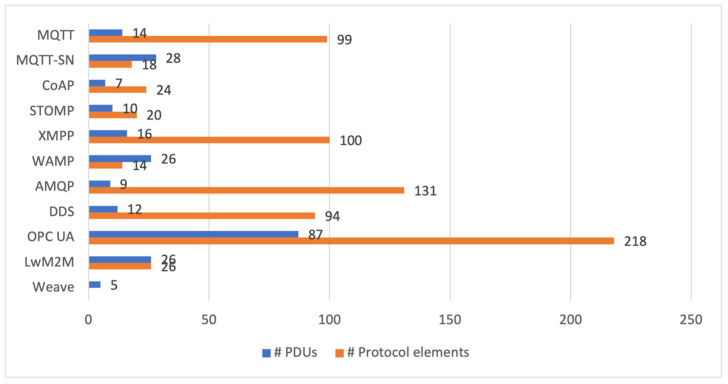
Complexity of the protocols (number of defined protocol elements).

**Table 1 sensors-21-06904-t001:** Protocol objectives and related features.

Protocol	Main Purpose	Defined Roles	Architectures	Discovery of	Real Time	Specific Features
MQTT	M2M & IoT small code footprint limited bandwidth and high latency networks	publisher, subscriber, broker	client-broker-client	topics	partial	
MQTT-SN	constrained devices	publisher, subscriber, broker	client-broker-client	topics, gateway	partial	
CoAP	constrained devices, lossy networks	server, client, support pub/sub mechanism	p2p, client-server, master-slave	resources	yes	RESTful
STOMP	simple data exchange	publisher, subscriber, server	client-m_server-client		partial	
XMPP	generalized routing of XML data	client, server	client-m_router-client	clients	limited	roster of user’s contacts & their presence status
WAMP	messaging & RPC	publisher, subscriber, broker, callee, caller, dealer	client-m_router-client, client-RPC_router-client	topics & RPCs	yes or partial	
AMQP	corporate environments	publisher, subscriber, broker	client-m_router-client, client-broker-client		partial	multiple links in one connection
DDS	real-time dependable systems, constrained devices (DDS-XRCE)	publisher, subscriber	p2p	publishers, subscribers, topics	yes	global data space, abstract API in IDL and mappings to C++ and Java
OPC UA	Industrial applications	client, server, publisher, subscriber	client-server, publishers- subscribers, servers-aggregator-clients, p2p,	application profiles, objects methods and variables	yes or partial	domain-specific information models
LwM2M	General M2M communication	client, server	client-server	devices	yes or partial	
Weave	smart-home	publisher, subscriber	p2p, client-broker-client			

**Table 2 sensors-21-06904-t002:** Messaging features.

Protocol	Payload Size Limits	Payload Data Representation	Labelling	Metadata	Transaction Support
MQTT	256 MiB	UTF-8 text, unspecified bytes	yes	yes	
MQTT-SN	64 KiB	unspecified bytes	yes		
CoAP	40 B–1 KiB (without IP fragmentation), 1 MiB–1 GiB with block-wise transfer	internet media type		yes	
STOMP	implementation dependent	internet media type	yes	yes	
XMPP	defined by end-points (64 KiB stanza size)	UTF-8 text	yes	yes	
WAMP	defined by end-points 512 B–16 MiB	JSON text and MessagePack	yes	yes	
AMQP	defined by end-points	internet media type	yes	yes	yes
DDS	64 KB 4 GiB with block-wise transfer	CDR, XML, user-def.	yes	yes	yes
OPC UA	defined by end-points	serialized binary data, JSON, XML	yes	yes	yes
LwM2M	depends on used transport and payload representation, for v. 1.0 mandatory TLV representation limit is 16.7 MiB	plain text, TLV, JSON, CBOR			
Weave	no strict limit advice to fit in transport protocol MTU	serialized binary data			

**Table 3 sensors-21-06904-t003:** Transport features.

Protocol	Basic [Additional] Transport	Security	QoS	Data Prioritisation	Addressing	Filtering Capability
MQTT	TCP, [WebSocket]	TLS profile: authentication +authorisation	3 levels		topic	
MQTT-SN	ZigBee or UDP		4 levels		topic	
CoAP	UDP, [TCP, WebSocket]	DTLS	2 levels		URI	URI syntax
STOMP	TCP, [WebSocket]				destination name	
XMPP	TCP, [WebSocket]	StartTLS, SASL		yes	Jabber Id.	
WAMP	WebSocket	realms, trust levels			URI	URI syntax
AMQP	TCP, [UDP, SCTP, WebSocket]	TLS, SASL	3 levels	yes	queue, topic/routing key	based on message properties
DDS	UDP, [TCP]	DDS Sec.: safety plug-ins	15 QoS politics 22 parameters	yes	topic/key	topic, time, content
OPC UA	TCP, WebSocket, HTTPS, UDP, [AMQP, MQTT]	Secure channel and security message fields		yes	URI, identifier from namespace	views, filter data structures
LwM2M	CoAP/UDP, CoAP/SMS, MQTT, HTTP	DTLS, dedicated objects			URI, ObjectID, Resource ID	
Weave	UDP, TCP	Security profile			IPv6 as resource ID, traits	

**Table 4 sensors-21-06904-t004:** Suitability for selected types of IoT devices.

Protocol	Constrained Devices	Payed Transmission	Temporary Off-Line	Always Online
MQTT	+/−	+	+	+
MQTT-SN	+	+	+	+
CoAP	+	+	+	+
STOMP	+/−	+	+	+
XMPP	−	+/−	+	+
WAMP	−	+/−	+	+
AMQP	−	+/−	+	+
DDS	+	+	+	+
OPC UA	+/−	+	+	+
LwM2M	+/−	+/−	+/−	+
Weave	+	+/−	+	+

**Table 5 sensors-21-06904-t005:** Suitability for basic communication purposes.

Protocol	Configuration	Data Acquisition	Data Querying	Alarms	Command Dispatching	Process Control	Direct Peer-to-Peer
MQTT	+	+	+/−	+	+	+/−	−
MQTT-SN	+	+	+/−	+	+	+/−	−
CoAP	+	+	+	+	+	+	+
STOMP	+	+	+/−	+	+	+	−
XMPP	+	+	+/−	+	+	+	−
WAMP	+	+	+	+	+	+	+/−
AMQP	+	+	+	+	+	+	+/−
DDS	+	+	+/−	+	+	+	+/−
OPC UA	+	+	+	+	+	+	+/−
LwM2M	+	+	+	+	+	+	+
Weave	+	+	+	+	+	+	+

## Data Availability

Not applicable.

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
