# Peer review of "Messaging Protocols for IoT Systems—A Pragmatic Comparison"

_sensors, 2021, doi:10.3390/s21206904_

Round 1

Reviewer 1 Report

This article describes and compares IoT protocols and is merely a survey paper on IoT protocols. It requires a good and thorough checkup on spellings and English usage.

Author Response

Response to Reviewer 1 Comments

Thank you for your comments.

Point 1: This article describes and compares IoT protocols and is merely a survey paper on IoT protocols. It requires a good and thorough checkup on spellings and English usage.

Response 1: Indeed we aimed to provide a comprehensive survey on IoT protocols based on a methodical analysis of the protocol specifications. It should be useful for IoT systems designers and researchers dealing with IoT communication aspects. Our observation is that many engineers and researchers working on IoT problems do not recognize all of the protocols described in the paper or do not understand their specificity. The language correctness of the article was carefully checked with the Grammarly tool and verified by an experienced author and reviewer of technical publications written in English.

Reviewer 2 Report

These points are missing in the study, although they might be important:
(1) Secure MQTT (SMQTT): a variant of MQTT, which uses attribute based encryption to allow broadcasting of encrypted message to multiple nodes. It allows symetric encryption where publishers and subscribers need to register themselves with the broker to obtain shared secret key.
(2) A brief duscussion/comparison between security and authentification mechanisms such as SSL and SASL is missing.
(3) Protocols (not presented in the study) that contain SASL support incude:
* LADP (Lightweight Directory Access Protocol)
* SMTP (Simple Message Transfer Protocol)
* IMAP (Internet Mail Access Protocol)
* POP3 (Post Office Protocol v3)
LADP, SMTP, IMAP, POP3 are also messaging protocols for IoT but are missing in the study.

Author Response

Response to Reviewer 2 Comments

Thank you very much for your constructive comments.

Point 1: These points are missing in the study, although they might be important:

(1) Secure MQTT (SMQTT): a variant of MQTT, which uses attribute based encryption to allow broadcasting of encrypted message to multiple nodes. It allows symetric encryption where publishers and subscribers need to register themselves with the broker to obtain shared secret key.

Response 1: We mentioned SMQTT on page 6, adding the following paragraph.

Some solutions to secure MQTT were proposed in the past. For example, in 2014, Neisse et al. [32] integrated their Model-based Security Toolkit with MQTT to support security and privacy requirements. In 2015, Singh et al. [33] defined a security extension for MQTT and MQTT-SN called Secure MQTT. The extension is based on the lightweight Elliptic Curve Cryptography and allows broadcasting encrypted messages to multiple nodes. However, the mentioned solutions were not included in the latest MQTT version from 2019, which introduces, among others, an enhanced authentication method. The method is commonly used to carry the SASL mechanism, but it is not constrained to SASL, and others like Kerberos can be applied. The MQTT specification strongly recommends using TLS for securing message exchange with the broker.

Point 2: (2) A brief duscussion/comparison between security and authentification mechanisms such as SSL and SASL is missing.

Response 2: We added a brief discussion on TLS and SASL on page 25, given in the following text.

Transport Layer Security (TLS) and Datagram Transport Layer Security (DTLS) protocols aim primarily to provide privacy and data integrity between communicating processes. They also allow authenticating one or two communication end-points. TLS works over TCP. DTLS is its functional equivalent that works over UDP. These protocols use public-key cryptography for secure key exchange. When the encryption key is securely exchanged, user data for efficiency purposes are encrypted using symmetric encryption, in most cases some variant of AES encryption. Moreover, data stored in the certificates could be used for the authentication of one side of communication. In most cases, in such a way server where clients are connecting is verified. However, TLS/DTLS introduces so-called mutual authentication during both communication sides are authenticated. In the IoT environment, this feature can be used for authentication of IoT devices as well as a central server, where they send data. TLS/DTLS can also be configured to work with pre-shared symmetric keys (RFC 4279). Thus, an IoT system designer can based authentication on digital certificates with public keys, raw asymmetric keys, and pre-shared symmetric keys.

Another alternative for providing security and authentication is using Simple Authentication and Security Layer (SASL) specified in RFC 4422. This protocol, called by authors – framework, allows the addition of security functions for currently used application protocols, like that used in the IoT environment. In contrast to TLS/DTLS, it contains an extensive list of possible authorization mechanisms, such as simple plaintext, various challenge-response mechanisms, and integration with Kerberos or OAuth. It could introduce data encryption and integrity checking; however, it allows the usage of TLS for this purpose. XMPP and AMQP could utilize both mechanisms – TLS and SASL at once.

StartTLS mechanism (aka. Opportunistic Encryption) defined in RFC 4322 allows starting an encrypted connection without any pre-arrangement specific to the pair of systems involved. StartTLS is used to initiate an encrypted connection on the same port as an unencrypted connection.  

Point 3: (3) Protocols (not presented in the study) that contain SASL support incude:

* LADP (Lightweight Directory Access Protocol)

* SMTP (Simple Message Transfer Protocol)

* IMAP (Internet Mail Access Protocol)

* POP3 (Post Office Protocol v3)

LADP, SMTP, IMAP, POP3 are also messaging protocols for IoT but are missing in the study.

Response 3: We mentioned the e-mail protocols on page 2, adding the following paragraph. However, we do not note LDAP, as it is a protocol for accessing and maintaining distributed directory information services. IoT devices and services can use it, but we cannot consider it a messaging protocol that carries measurement or control information between IoT devices and services.

An IoT system designer can use e-mail protocols (e.g. SMTP, IMAP, POP3) for message exchange; he can use even a social-media as a transport layer platform (e.g. Twitter). We decided to exclude them from the comparison. The principal aim of e-mail is to serve a human at least at the recipient end of the communication. Moreover, today’s e-mail services are overwhelmed by spam. We cannot consider them as a recommended solution for communication between IoT devices and services. A solution based on a social-media can be considered as a simple transport without specific functionalities expected in IoT communication.

Reviewer 3 Report

  • Lack of contributions: The paper does not provide any novel solution, technical contributions, or a deep technical discussion on the messaging protocols. The provided comparisons and suggestions stay on surface and only provide a selection guide for engineers.
  • Lack of deep comparison: Since the manuscript reviews the existing methods, it is expected to fulfill this comparison in a thorough way. More graphs and quantifications (along side with the tables) could be helpful to give a better idea to the readers about the parameters which are mentioned in the table. Many of the comparisons in the tables are qualitative by using words while numbers and graphs can help the reader grasp the pros and cons better. The three figures in the paper are preliminary and, in some ways, trivial.
  • Incomplete list of references: only 31 references for such a topic is insufficient.
  • The paper tries to be comprehensive in providing some suggestions, but there is no clear path in the suggestions and in practice will be confusing and hard to follow. One of the reasons of this confusion is the choice of parameters which is sometimes trivial so all the protocols own that specific property, and it is not clear quantitatively how it works in each one of them. Appropriate references have not been provided for the further study and investigation of the reader and the paper does not provide any vision for future protocols for researchers to investigate.

Author Response

Response to Reviewer 3 Comments

Thank you very much for your comments. They helped us to add some explanations to our article. 

Point 1: Lack of contributions: The paper does not provide any novel solution, technical contributions, or a deep technical discussion on the messaging protocols. The provided comparisons and suggestions stay on surface and only provide a selection guide for engineers.

Response 1: The paper, being a comparison of IoT messaging protocols, does not provide any novel solution. The comparison is based on the specifications of the protocols. It covers all messaging protocols used in IoT systems, in contrast to many existing comparisons that compare a few of them based on selected implementations. Our observation is that many engineers and researchers dealing with IoT communication do not recognise all of the protocols described in the paper. Analysing existing comparisons, we notice confusing the features of protocols with the properties of their selected implementations. The implementation-related properties probably seem more technically deep, but they should not lead to wrong assessments of the protocols. Different implementations of the same protocol and diverse application scenarios make the existing analysis incomparable. We hope that our paper will be a useful guide for IoT engineers and a valuable introduction for researchers in the IoT communication domain.

Point 2: Lack of deep comparison: Since the manuscript reviews the existing methods, it is expected to fulfill this comparison in a thorough way. More graphs and quantifications (along side with the tables) could be helpful to give a better idea to the readers about the parameters which are mentioned in the table. Many of the comparisons in the tables are qualitative by using words while numbers and graphs can help the reader grasp the pros and cons better. The three figures in the paper are preliminary and, in some ways, trivial.

Response 2: We added the following two paragraphs on that issue on page 32 and 33 respectively.

We tried to gather the performance measurements from different articles and put them together, but such data was misleading. The published results are incomparable. They are related to different usage scenarios and software and hardware platforms applied in the performed tests.

The presented comparison is more qualitative than quantitative. It is not easy to find qualitative parameters for comparing protocols basing on their specifications. Even simple parameters like min/average/max header and message sizes depend on application scenarios and selected protocol options, and we failed to calculate them. The interesting parameters like processor, memory, bandwidth consumptions, effective-to-transmitted bitrate are implementation-dependent. To make a credible protocols comparison based on their implementations, a team should build at least three different IoT communication scenarios based on at least three different implementations of each protocol on the same hardware devices. Probably it is a future challenging work to do, but it is huge work. It is also challenging to define some benchmarks for such comparison. 

Point 3: Incomplete list of references: only 31 references for such a topic is insufficient.

Response 3: Our work is based on the protocols specifications, which are numerous, and we decided not to include all of them in the bibliography. Adding them would raise the references number to over 100 items. We do not consider it gives any value to the reader. Our references point to documents that we consider valuable, and we can recommend for the reader. They are mainly related works we shortly describe in Section 6. Our paper is not an analysis of existing publications about messaging protocols for IoT, thus in our opinion, the number of references is adequate to the paper content.

Point 4: The paper tries to be comprehensive in providing some suggestions, but there is no clear path in the suggestions and in practice will be confusing and hard to follow. One of the reasons of this confusion is the choice of parameters which is sometimes trivial so all the protocols own that specific property, and it is not clear quantitatively how it works in each one of them.

Response 4: We have to admit that some of the analysed properties are the same for almost all protocols. It is the result of the continuous evolution of the protocols and that communities related to the protocols tend to make them universal and commonly used. Consequently, the choice of the suitable protocol is difficult, and there are no straight rules for their selection. We added the following paragraph on that issue on page 32. 

The communities engaged in particular protocol evolution tend to make the protocol universal and commonly used. Thus, the results of suitability and applicability analyses (Table 4 and 5) do not allow for easy protocol classification. Consequently, the choice of a suitable protocol for a given application is more complicated.

Point 5: Appropriate references have not been provided for the further study and investigation of the reader and the paper does not provide any vision for future protocols for researchers to investigate.

Response 5: The best selection for further study is the protocols specifications if someone needs to learn more about them. A selection of other papers would be specific for the particular needs of the reader. The vision for future research is given in the last statement of the paper, i.e. "Thus, there is a need for a comprehensive analysis and comparison of IoT data and metadata representation – what point an important direction for further research." 

Round 2

Reviewer 3 Report

  • I would suggest clarify the point you mentioned in the response letter  regarding the utilized methodology in your comparisons in the paper. 
  • Check the grammar and spelling.
